

# 1  Use of automatic radiosonde launchers to measure temperature and humidity
# 2  profiles from the GRUAN perspective

Fabio Madonna[1], Rigel Kivi[2], Jean-Charles Dupont[3], Bruce Ingleby[4], Masatomo Fujiwara[5], Gonzague
Romanens[6], Miguel Hernandez[7], Xavier Calbet[7], Marco Rosoldi[1], Aldo Giunta[1], Tomi Karppinen[2],
Masami Iwabuchi[8], Shunsuke Hoshino[9], Christoph von Rohden[10], Peter William Thorne[11]
[1]Consiglio Nazionale delle Ricerche - Istituto di Metodologie per l'Analisi Ambientale (CNR-IMAA), Tito Scalo (Potenza), Italy
[2]Finnish Meteorological Institute, Helsinki, Finland
[3]Institut Pierre et Simon Laplace (IPSL), Paris, France
[4]European Centre for Medium-range Weather Forecasts (ECWMF), Reading, UK
[5]Hokkaido University, Sapporo, Japan
[6]MeteoSwiss, Payerne, Switzerland
[7]Agencia Estatal de Meteorología, Madrid, Spain
[8]Japan Meteorological Agency (JMA), Tokyo, Japan.
[9]Aerological Observatory, Tsukuba, Ibaraki, Japan.
[10]Deutscher Wetterdienst (DWD), GRUAN Lead Centre, Lindenberg, Germany.
[11]Irish Climate Analysis and Research Units, Dept. of Geography, Maynooth University, Maynooth, Ireland.

## 20  Abstract

In the last two decades, technological progress has not only seen improvements to the quality of
atmospheric upper-air observations, but also provided the opportunity to design and implement
automated systems able to replace measurement procedures typically performed manually.
Radiosoundings, which remain one of the primary data sources for weather and climate
applications, are still largely performed around the world manually, although increasingly fully
automated upper-air observations are used, from urban areas to the remotest locations, which
minimise operating costs and challenges in performing radiosounding launches. This analysis
presents a first step to demonstrating the reliability of the Automatic Radiosonde Launchers (ARLs)
provided by Vaisala, Meteomodem and Meisei. The metadata and datasets collected by a few
existing ARLs operated by GRUAN certified or candidate sites (Sondakyla, Payerne, Trappes,
Potenza) have been investigated and a comparative analysis of the technical performance (i.e.
manual vs ARL) is reported. The performance of ARLs is evaluated as being similar or superior to
those achieved with the traditional manual launches in terms of percentage of successful launches,
balloon burst and ascent speed. For both temperature and relative humidity, the ground check
comparisons showed a negative bias of a few tenths of a degree and % RH, respectively. Two
datasets of parallel soundings between manual and ARL-based measurements, using identical sonde
models, provided by Sodankylä and Faa'a stations showed mean differences between the ARL and
manual launches smaller than ±0.2 K up to 10 hPa for the temperature profiles. For relative
humidity, differences were smaller than 1% RH for the Sodankylä dataset up to 300 hPa, while they
were smaller than 0.7% RH for Faa'a station. Finally, the O-B mean and rms statistics for German





RS92 and RS41 stations which operate a mix of manual and ARL launch protocols, calculated using
the ECMWF forecast model, are very similar, although RS41 shows larger rms(O-B) differences for
ARL stations, in particular for temperature and wind. A discussion on the potential next steps
proposed by GRUAN community and other parties is provided, with the aim to lay the basis for the
elaboration of a strategy to fully demonstrate the value of ARLs and guarantee that the provided
products are traceable and suitable for the creation of GRUAN data products.
**1.    Introduction**
Radiosondes are one of the primary sources of upper-air data for weather and climate monitoring.
Despite the advent and the fast integration of GPS-RO (radio occultation) as an effective source of
upper-air temperature data (Ho et al., 2017), radiosondes will likely remain an indispensable source
of free-atmosphere observational data into the future. Radiosonde observations are applied to a
broad spectrum of applications, being input data for weather prediction models and global
reanalysis, nowcasting, pollution and radiative transfer models, monitoring data for weather and
climate change research, and ground reference for satellite and also for other in-situ and remote
sensing profiling data.
The analysis of historical radiosonde data archives has repeatedly highlighted that changes in
operational radiosondes introduce clear discontinuities in the collected time series (Thorne et al.,
2005; Sherwood et al., 2008; Haimberger et al., 2011). Moreover, where radiosonde observations
have been used in numerical weather prediction, systematic errors have sometimes been
disregarded and the instrumental uncertainties have been estimated in a non-rigorous way
(Carminati et al., 2019). Nowadays, there is a broad consensus on the need to have reference
measurements with quantified traceable uncertainties for scientific and user-oriented applications.
The GCOS Reference upper-air network (GRUAN) provides fundamental guidelines for establishing
and maintaining reference-quality atmospheric observations which are based on principal concepts
of metrology, in particular, traceability (Bodeker et al., 2016).
Apart from direct instrument performance aspects of the radiosounding equipment and radiosonde
model, it must be acknowledged that there are many challenges in performing radiosounding
launches. During the preparation and launch phase, many circumstances may interfere with the
smooth operation of radiosoundings such as undertaking launches at night, harsh meteorological
conditions for balloon train preparation compounded by basic equipment in the balloon shelter, if
any, and safe handling when using hydrogen as balloon gas, and last but not least the risk of
errors/mishandling by the operators. Additional expenditure may be required when observations



are performed in remote regions of the globe, including the polar regions, deserts, or remote
islands.
Since the start of radiosounding efforts in the early-to-mid 20th Century, the radiosounding systems
and the radiosondes themselves have radically changed in size, weight, performance. For example,
a very important progress was the automation of the data processing and message production from
about 1980. Of particular note is that thanks to new technologies, over recent decades, three
manufacturers have developed and deployed fully Automatic Radiosonde Launchers (ARL) able to
perform unmanned soundings.
ARL are robotic systems able to complete in an automatic fashion almost all of the operations
performed manually by an operator during radiosounding launch preparation and release, including
the implementation of ground check procedures. The advantages of ARLs are in the reduction of
the challenges described above as well as in the reduced running costs of a sounding station (e.g.
reduction in the need for trained staff and the trend of automating hydrogen production due to cost
reasons and to the helium international crisis) and in ameliorating problems of recruiting long-term
operators for remote locations. Nevertheless, it must be also stressed that the system must be
regularly stocked and maintained to avoid major issues and high repair costs being incurred. In
addition, with changes in the radiosonde technology, updates of the systems might be required to
enable the use of a new radiosonde type, with periodical costs (variable, every 3-6 years) which
might be substantial for a station. In 2018, NOAA-NCEI published stories on its website which show
the potential benefits of using ARLs (http://www.noaa.gov/stories/up-up-and-away-6-benefits-of-
automated-weather-balloon-launches). Within these stories as well as from the feedback collected
within the GRUAN community, several radiosonde stations have reported benefits from the use of
ARL and an increase in the percentage of successful soundings with a potential reduction of missing
data in the collected data records.
Using recent ECMWF statistics on the number of stations transmitting data to the WMO Information
System (WIS) and information provided by the GRUAN community and others, there are about 90
ARLs (Figure 1) providing data versus about 700 manual stations. ARL stations cover many countries
and remote regions, including Arctic and Antarctic locations as well as a broad suite of remote Pacific
and other island locations. As far as is known many of the ARL stations only make automated
launches. In addition, there are a few more stations, used by research institutions or environmental
agencies, not transmitting data via the Global Telecommunication System (GTS) of the WMO


Information System (WIS). The total number of stations operating an ARL worldwide has increased
within the last decade (see Table A1 and A2 in Appendix A).
Vaisala introduced its first automatic system in 1990, Meisei in 2006 and Meteomodem in 2009.
Despite their relatively recent development and deployment, ARLs appear to be successful, and the
number of deployed systems will likely increase in the future. However, to date there are very few
peer-reviewed papers in the literature dealing with ARLs or comparing ARL vs manual data (often
limited to specific examples, e.g. Madonna et al., 2011). More specifically, there is currently no side-
by-side assessment of quality in comparison to manually launched sondes. The aim of this paper is
thus to quantify the reliability and stability of ARLs and assess the accuracy of their data compared
to the traditional manual systems. A discussion on the measurement traceability and on the
feasibility to use ARLs in a regular way in the GCOS Reference Upper Air Network (www.gruan.org)
is also provided. At present, traceability to SI standards is quantified at several GRUAN sites by the
use of a Standard Humidity Chamber (SHC) which can be used for ARL before the launch loading
only. The SHC is a simple ventilated chamber (~4 – 5 m/s) using distilled water which, during the
ground check procedure, is first heated a few degrees above ambient temperature and then cooled
to saturate air at 100% relative humidity. The SHC allows a check of each radiosonde at 100% RH
using distilled water (or other RH values using solutions with specific salts although these are
generally only used at the GRUAN Lead Centre and for sonde characterisation and not operational
sounding preparation purposes).
The comparison reported in this paper focuses exclusively on temperature and relative humidity
profiles and rely upon manufacturer's products (i.e. GRUAN Data Processing based on the raw data
collected by the sonde, described in Dirksen et al., 2014, and Kobayashi et al., 2019, is not used).
The remainder of the paper is structured as follows. In section 2, a short description of the three
ARLs is provided. In section 3, the technical performance of the ARLs is investigated on the basis of
statistics comparing the technical efficiency of the ARLs versus the manual sounding stations as well
as reporting an analysis of the feedback from station operators collected at the GRUAN sites on the
advantages, limitations and technical issues faced to maintain and ensure continuity of ARL
operations. Section 4 reports on the effect of the usage of ARLs on the stability and the accuracy of
ground-check calibration procedures. Section 5 provides statistics obtained from parallel soundings
at different sites for both temperature and humidity profiles. Section 6 discusses the comparison
between observation-minus-background (O-B) statistics obtained from ARL data and manually
launched data, respectively, using the ECMWF short-range forecast fields. Finally, section 7 provides



a summary and a description of the experiments which might be performed to design future ARL
setup to enable full measurement system traceability to SI units and, therefore, to meet GRUAN
requirements for long term reference climate data.

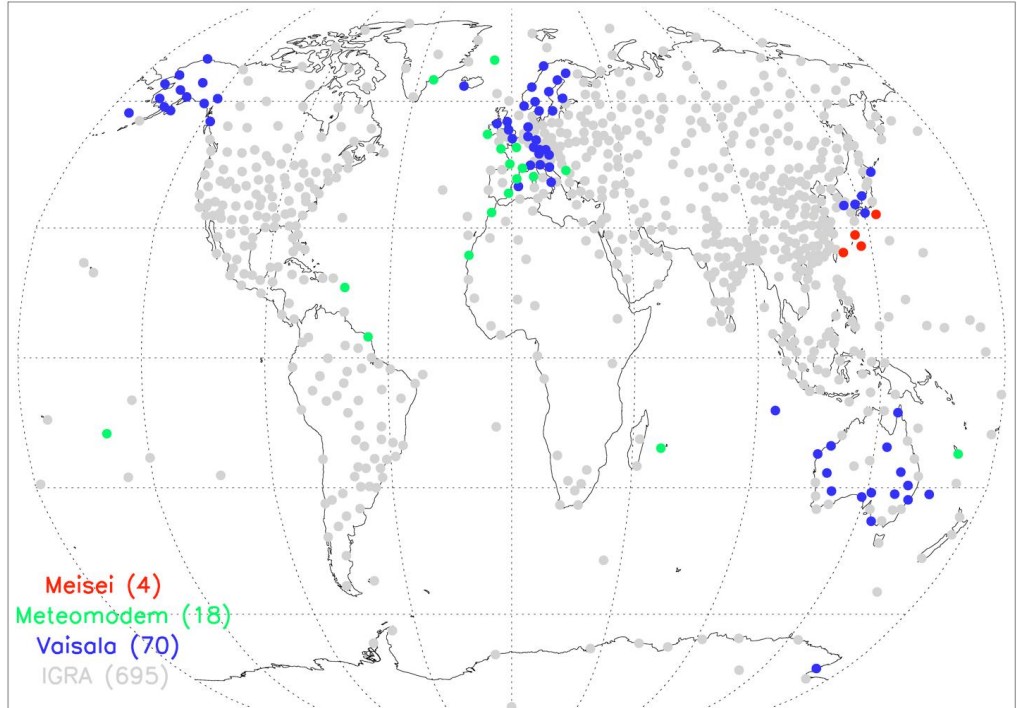


Figure 1: Map of stations running an Automatic Radiosonde Launcher (ARL) and transmitting the data to the WIS in late
2019 (see also Appendix A). Blue dots are the Vaisala ARL, green the Meteomodem, and red the Meisei. In light grey,
the manual station providing data to the WIS in September are also reported. Number of stations for each color is
reported in brackets.
**2.   Description of existing ARL systems**
**2.1 Vaisala Autosonde: brief history and recent system configurations**
Automation of upper-air sounding data processing has made steady progress since the early 1970's
and is now widespread (Kostamo, P., 1992). The Vaisala Autosonde project was started in late 1992
and a working prototype presented at CIMO, Vienna, in 1993. The prototype was tested in Norway
and Sweden in 1993 and 1994. This coincided with the replacement of manual balloon tracking
systems by Omega and Loran networks. It was provided by Vaisala Oy (Finland) and was installed at
the Landvetter station in Sweden in 1994. As of today, about 80 Vaisala ARLs have been installed



worldwide and the number of soundings performed has exceeded 800,000, while the annual
number of new soundings will soon exceed 70,000 (Lilja et al., 2018). With the newest Autosonde
model it is possible to perform 60 soundings without replenishment, while the earlier models
allowed up to 24 soundings.
The first radiosonde type used for an automatic launch was the RS80-15N (during 1994-2006). The
RS80 radiosonde was followed by the models RS92 (manufactured 2005-2017) and then RS41
(available since late 2013). The RS92 radiosonde (Dirksen et al. 2014) which performs
measurements with a nominal measurement uncertainty (provided by the manufacturer) of 0.5°C
for temperature, 1.0 hPa for pressure below 100 hPa and 0.6 hPa above, 0.15 m s$^{-1}$ for wind speed
and 5 % RH or relative humidity (https://www.vaisala.com/sites/default/files/documents/RS92SGP-
Datasheet-B210358EN-F-LOW.pdf). RS41 sonde specifications for nominal measurement
uncertainties (provided by the manufacturer) are 0.3°C for temperatures below 16 km and 0.4°C
above, 0.01 hPa for pressure sensor, 0.15 m s$^{-1}$ for wind speed and 4 % RH for relative humidity
(https://www.vaisala.com/sites/default/files/documents/RS41-SGP-Datasheet-B211444EN.pdf).
Note that the Vaisala RS41 radiosondes are of two different types: RS41-SG which are not equipped
with a pressure sensor and using the GPS-based method to infer pressure (Lehtinen, 2014), and
RS41-SGP which uses a pressure sensor as the default. More stations use the RS41-SGP than the
RS41-SG: in November 2019, 158 stations type RS41-SGP versus 66 stations using type RS41-SGP.
To launch the RS41 sondes, the Autosonde Ground Check (GC) procedure has been updated. The
GC device of the RS41 sondes consists of a wall-mounted box and an activator that contains a
wireless reader for the radiosonde. The device is designed to automatically activate the radiosonde
and to enable wireless data transfer. An activator is connected to the reader box with a coaxial
cable. The ground check device also includes a barometer while the surface pressure used as a
reference for the launch is obtained from a separate co-located automatic weather station.
However, the ground check pressure device can be used as a backup for the weather station sensor.
The GC performs a temperature check where the actual temperature sensor is compared with the
one integrated on the humidity sensor chip. In contrast to the RS92 GC, a pre-flight fine-tuning of
the temperature measurement is no longer applied to the RS41 because the manufacturer found
that the accuracy of the RS41 temperature measurement is practically unchanged during storage.
Humidity is also checked in the GC. The RS41 humidity check consists of two main steps – the sensor
reconditioning phase and the 0% RH check. In the reconditioning phase, the sensor is heated to
remove possible contaminants that might affect the measurement results and cause a slight



degradation of the sensitivity of the humidity sensor. Then, the humidity sensor is checked and then
corrected against a dry humidity condition. Specifically, the dry reference condition of the new zero
humidity check is generated in open air by heating the sensor using the integrated heating element
on the sensor chip. The procedure is based on the decrease of relative humidity towards zero as the
temperature rises high enough. This method differs from the RS92 GC where the correction was
based on a dry condition generated with desiccants, whose drying capacity gradually fades with the
time.
The radiosonde's humidity sensor is reconditioned and ground check performed during the
automated launch preparation in order to ensure same performance as in manual stations (Lilja et
al., 2018). The top panel of Figure 2 provides a schematic picture of the most recent VAISALA AS41
Autosonde system configuration while the bottom panel shows a photograph of the Autosonde
system operational at the Finnish Meteorological Institute GRUAN site in Sodankylä (WIGOS station
identifier=0-20000-0-02836, 67.34 °N, 26.63 °E, 179 m a.s.l.). In Table 1, the basic technical data of
the Autosonde AS41 are reported. More details on the specifications of the Vaisala Autosonde AS41
can be found in the datasheet (B211636EN-A_2 pages.pdf, last accessed September 20, 2019)
available on the Vaisala website (https://www.vaisala.com).






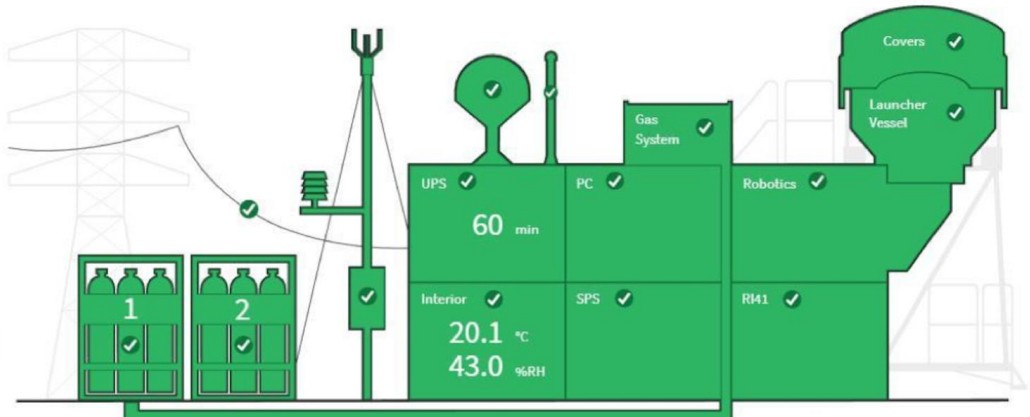

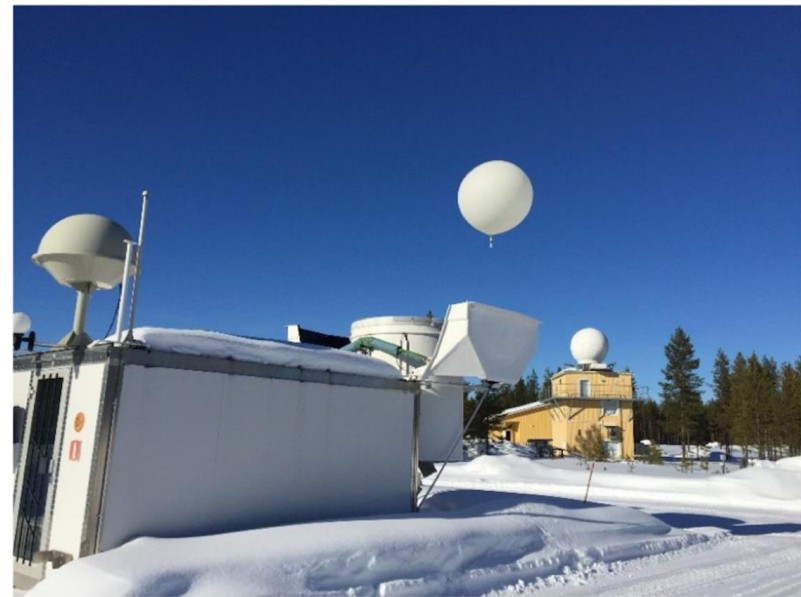



Figure 2: Schematics of the VAISALA Autosonde AS41 system in its most recent configuration (top panel), and photo of
the Autosonde system AS15 (bottom panel) operational at the Finnish Meteorological Institute GRUAN site in Sodankylä
(WIGOS station identifier=0-20000-0-02836, 67.34 °N, 26.63 °E, 179 m a.s.l., see Vaisala 2018,
https://www.vaisala.com/sites/default/files/documents/AUTOSONDE%20AS41%20Datasheet%20B211636EN-
A_2%20pages.pdf)).








217                               Table 1: Autosonde AS41 technical data (Vaisala, 2018)

| Dimensions | Width: 3.30 m |
|---|---|
| | Length: 7.80 m |
| Launch Tube Diameter | 2.20 m |
| Height during transport | 2.90 m |
| Total height with launcher tube | 5.10 m |
| Gross weight with launcher tube | 7.5 t |
| Electrical energy consumption | < 1 kW (without air conditioning) |



**2.2 Meteomodem Robotsonde**

The Meteomodem ARL is an automatic balloon launcher system that can perform up to 12 or 24
soundings without any manual control (http://www.Meteomodem.com/docs/en/Leaflet-
robotsonde.pdf). The system is compatible with M10 and M20 Meteomodem radiosonde types. It
is built in a robust dry maritime container and composed of the following subsystems (Figure 3):
● Operator room with electronic control unit and PC workstation, isolated from the launch tube
by an air-tight safety door, and used only during radiosonde setup and restocking;
● Carrousel with 12 or 24 removable containers for balloon trains, and with individual flexible
cover on balloon locations which preserve balloons from desiccation;
● Launch tube for balloon inflation and release and pneumatic equipment or pressurized air
network;
● Optionally, a double-door entrance to protect from strong winds, rain, drifting snow or
sandstorms.
The Meteomodem ARL main specifications are reported in Table 2. Worldwide there are 19
Meteomodem ARL systems automatically launching Meteomodem M10 radiosondes. The
specifications for nominal measurement uncertainties (provided by the manufacturer) are 0.58°C
for temperature, 1 hPa for pressure, 0.15 m s$^{-1}$ for wind speed and 5 % RH for relative humidity
(www.Meteomodem.com/docs/en/Leaflet-m10.pdf).




Table 2: Meteomodem ARL specifications

| Dimensions | Width: 2.44 m |
| --- | --- |
| | Length: 6.00 m |
| Launch Tube Diameter | 2.00 m |
| Height during transport | 3.10 m |
| Total height with launcher tube | 3.60 m |
| Gross weight with launcher tube | 3.5 t |
| Electrical energy consumption | < 1 kW (without air conditioning) |

For each launch, there is a preparation phase which comprises the radiosonde GC and the loading
of the balloon train (with the radiosonde, the unwinder, the parachute, and the balloon) into
individual bins before finally sounding parameters (e.g. launch time schedule, inflation volume, etc.)
are setup.
During the launch phase, before powering on the sonde, the system performs a scan of the
bandwidth in order to detect possible radio interference, then the radiosonde battery pack is
powered on through an infrared link. According to the scan result, the system sets up the new
frequency through an infrared link, and GNSS signal collection is initialized. Then, the system loads
the calibration data of the relevant radiosonde stored during the preparation phase and checks
consistency with PTU criteria. The Meteomodem ARL GC is a standard Meteomodem GC which
consists in a sealed box enclosing a reference and a fan which homogenised the inside temperature
and relative humidity. It is recommended to return the Meteomodem GC every 3 years for
calibration. The calibration is made with a certified Rotronic HC2A-S probe.
Then, the ARL records the ground check data and the metadata. Balloon inflation starts accordingly:
the system monitors a flowmeter to inflate the balloon to the specified volume. The ARL may use
either helium or hydrogen gas. Finally, the balloon is released at the specified launch time. In case
of launch failure before balloon release or during the flight, the procedure will restart for a new
sounding immediately or can alternatively be manually launched according to a preset time
schedule. At any time, an immediate start of the launch procedure can be initiated by an operator
(locally or remotely).





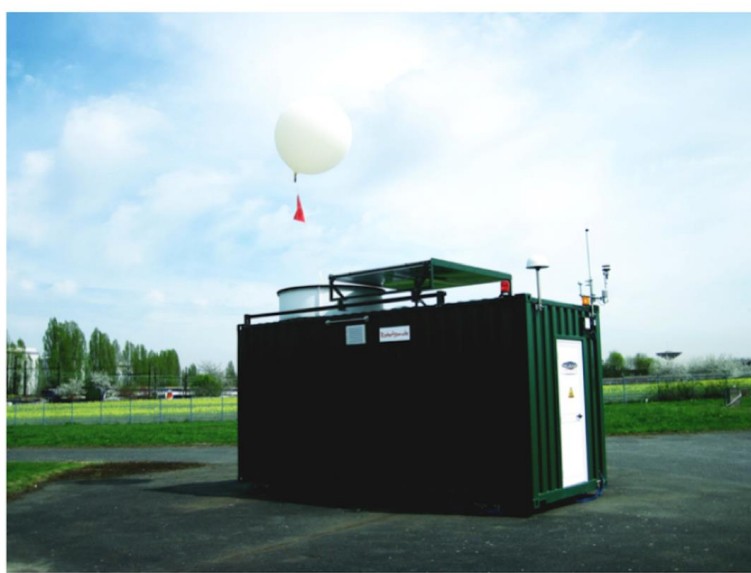

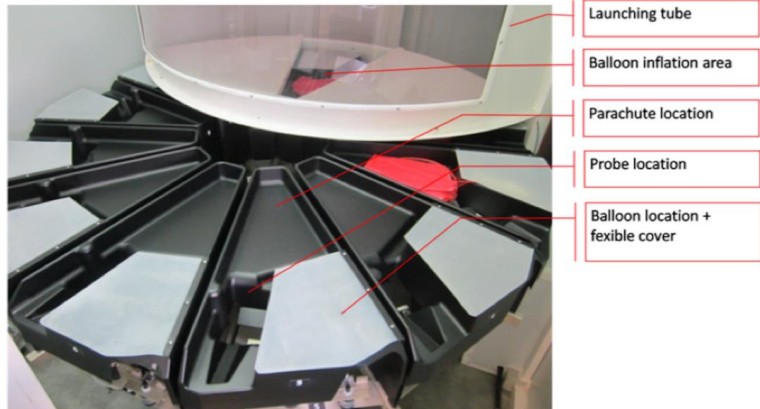

Figure 3: Meteomodem Robotsonde (top panel) launching a balloon at Trappes station (WIGOS station identifier=0-20000-0-07145, 48.46N, 0.20E, 168 m asl,  http://www.Meteomodem.com/robotsonde.html) and photograph of the carousel of Meteomodem Robotsonde with the balloon location (bottom panel).

For those stations operating an ARL and adopting a protocol based on GRUAN recommendations (Dirksen et al., 2014), as at Trappes station (WIGOS station identifier=0-20000-0-07145, 48.46N, 0.20E, 168 m asl, top panel of Figure 2.2), the GRUAN M10 ground check procedure is performed in two steps: 5 minutes in a ventilated hut in ambient conditions together with calibrated T and RH sensors and, further, another 5 minutes to test the radiosonde performance in the SHC. Then each radiosonde is loaded in the ARL carousel (bottom panel of Figure 3).



A technical document describing the M10 sensor, corrections and uncertainties for both the
temperature and relative humidity sensors will become available through the GRUAN community
as soon as a Meteomodem M10 GRUAN data product is available.

**2.3 Meisei Automated Radiosonde System**
The Meisei ARL, named "Automated Radiosonde System" is designed for fail-safe operation and
high remote operability. Compared to the previous version developed in 2006, the new system is
able to load more radiosondes thanks to the development of the Meisei "Canister Type". The
operator can preload a maximum number of 40 sondes adjustable in the so-called "Canister
modules". The canister has been recently implemented to reduce failures. Once the launch
procedure has started, the respective canister fills a balloon independently. The right canister
module and the left canister module are independent systems. It realizes high observation
continuity by duplicating gas, air and electric systems. The canister module on one side can be
moved to the preparation room to load the sonde and facilitate the operator's work. The new ARL
version can also recover from balloon bursts without human intervention at the site by using a
balloon from another canister. In the previous version, an operator had to visit the ARL to remove
broken balloons and restart the ARL during the observation window in such cases.
The new system is also equipped with a new simplified wind shield for launches in strong wind
conditions. All information and data are stored in a database available for each ARL. Various central
monitoring/control functions are provided by using application software and a web browser to
access the database on the workstation installed in the ARL. The Meisei ARL GC consists of a
temperature and humidity reference sensor and an inspection box. he GC performs before the
sonde loading. The results from the GC are not used in the data processing but only to check if there
are anomalies in the radiosondes.
In Table 3, the Meisei Automated Radiosonde System specifications are provided.
Figure 4 shows a photo of the system along with a sketch of the internals of system container. For
more details on the Meisei ARL experimental setup visit the Meisei website
(http://www.meisei.jp/ars). Japan Meteorological Agency (JMA) collects Meisei ARLs data since
2006. Parallel radiosoundings of auto launch and manual launch have not been done yet. This is the
reason why this paper does not show additional datasets or comparisons involving Meisei ARL: at
this stage, the description of the Meisei ARL is the only information which can be shared with
readers, according to recommendations provided by Meisei.





Table 3: Meisei ARS specifications

| Dimensions | Width: 2.50 m |
|---|---|
| | Length: 6.20 m |
| Launch Tube Diameter | 2.20 m x 1.80 m square |
| Height during transport | 3.10 m |
| Total height with launcher tube | 1.90 m (2.80 m including windshield) |
| Gross weight with launcher tube | 6 t |
| Electrical energy consumption | < 1 kW (without air conditioning) |


**3.    Technical performance**


Beyond the automation of the radiosonde launch procedure, there are two main differences between an ARL and a manual launch:

- Ground check procedures may be performed only during the sonde loading in the carrousel chamber, days or weeks before the sonde launch, though there is a trend towards less frequent stocking;
- The use of independent and traceable calibration standards like the Standard Humidity Chamber (SHC) is possible but only before the launch loading (also in this case one or more days before the launch).

Both these aspects will be discussed in the following sections which provide potential technical solutions to address the gaps between manual and automatic launch procedures in terms of performance and traceability.




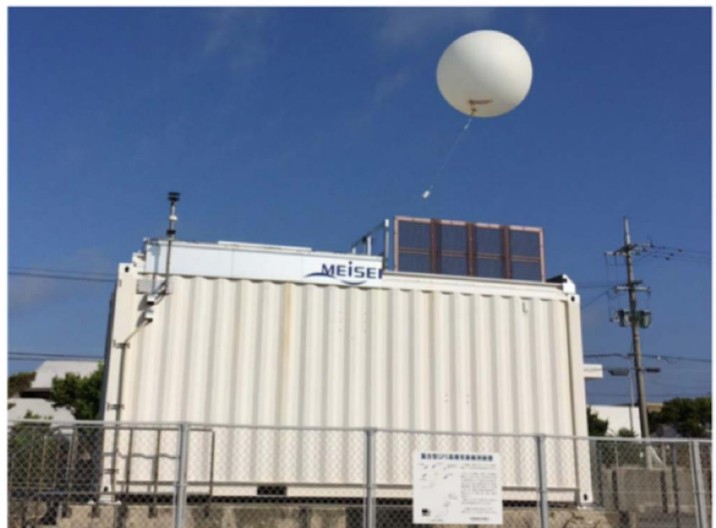

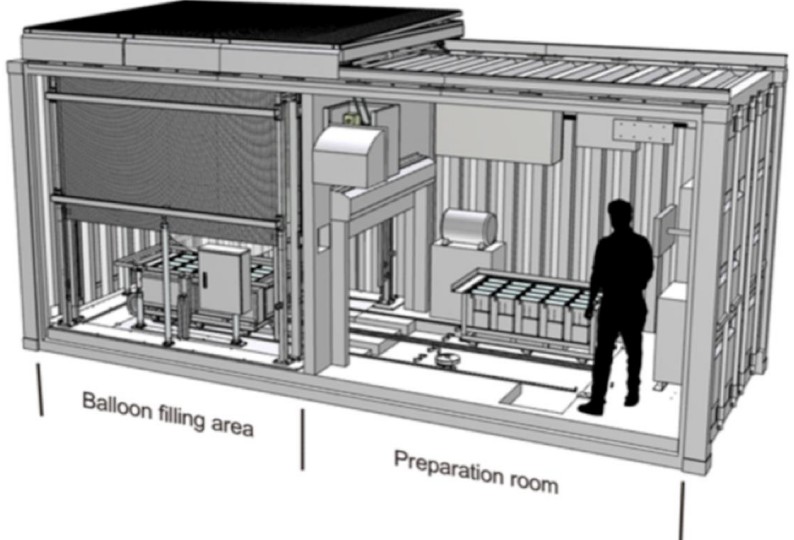


Figure 4: Picture of a Meisei Automatic Balloon Launcher (top panel) and sketch of the internals of ARL container in its
most updated configuration (bottom panel).


This section instead aims to provide a classification of the main challenges met by the stations which
have operated ARLs over several years and to assess the technical performance of the ARLs
compared to manual launches. The section is built upon the feedback provided by the GRUAN sites
in response to a survey for the collection of ARL information. Most of the ARLs at GRUAN sites are
from Vaisala (thus the analysis is not representative of Meisei and Meteomodem systems due to
the very limited feedback available for these systems). Given the small sample size, this is presented



qualitatively rather than quantitatively and it is anonymised. Examples of technical performance in
the field are then provided for a Vaisala and a Meteomodem ARL operating the most recent updated
version of the respective manufactured systems (at Payerne and Trappes stations).
A conceptual diagram to represent a generic ARL is provided in Figure 5: each ARL can be
schematically divided into 4 areas as follows:
● the operator's area, where the operators can manage the system, prepare radiosondes and

balloons to be uploaded and where the station reception and processing units are located;

● the ready-to-launch sondes storage area, built around the ARL rotating trays, where most

of the automated technologies are implemented to allow a completely unmanned launch;

● the launching vessel area, where the balloon is filled and becomes ready for the launch;
● external area, where all the ancillary instruments, such as the weather station and GNSS

antenna, are located along with gas tanks.

For each area, the weakest points identified from the GRUAN sites operating an ARL are:
● in the operator's area, most of the issues are related to the not infrequent failure of power

supply system or of the air conditioning system, often related to a major failure of the power

supply at the measurement station itself; this anyhow represents a weakness in the use of

ARLs in remote areas, where logically the ARL might be an obvious choice; a few sites also

reported issues in the software and logic controllers;

● the ready-to-launch sonde storage area is surely the most efficient part of ARLs, where few

issues are reported which indicates the robustness of these areas; the most critical issue

identified in this area is the infrequent failure of the air compressor;

● the launching vessel area is where the balloon is filled and launched and where, therefore,

we have a high exposure to many environmental factors like harsh climate, dust, animals,

etc., which can strongly affect a successful launch also with later effects to the balloon and

early burst; several issues are raised by the stations related to challenges in the balloon

inflation process, failure of balloon presence sensor allowing launch of under-inflated

balloons, gas tubes bent and frozen gas hoses, balloon blocked on the tray, failure of the

rams which open vessel cover doors (this concerns Vaisala or Meisei, and not Meteomodem

ARL), delays in launch detection time compared to the actual launch time, occasional break

of the radiosonde string at launch (for Meisei);

● the external area, is another critical area where several problems have been reported about

the gas flow meter and the switching between the gas tanks (one close to empty and the





other fully filled); extreme weather conditions (e.g very strong winds) can make the launch
more difficult, despite the additional screens protecting the balloon flight in the first 2-3
meters above the ARL (only for Vaisala and Meisei).
The problems listed above are not common to all the ARLs, each system has its own specific issues.
On one side, the feedback reported from GRUAN stations can provide a first assessment of the
challenges in operating an ARL: this study cannot assess challenges in the operation of each specific
model and it cannot quantify the improvements of each ARL with the time. The issues discussed
above could be used as recommendations to the manufacturers to foster further improvements of
the systems. The ARLs are typically maintained by the manufacturers on an annual check up
(performed remotely) and major maintenance approximately every 3 years. This maintenance
schedule, if applied at each station can increase the reliability of the systems over short and long
term, although it generates a cost increase.

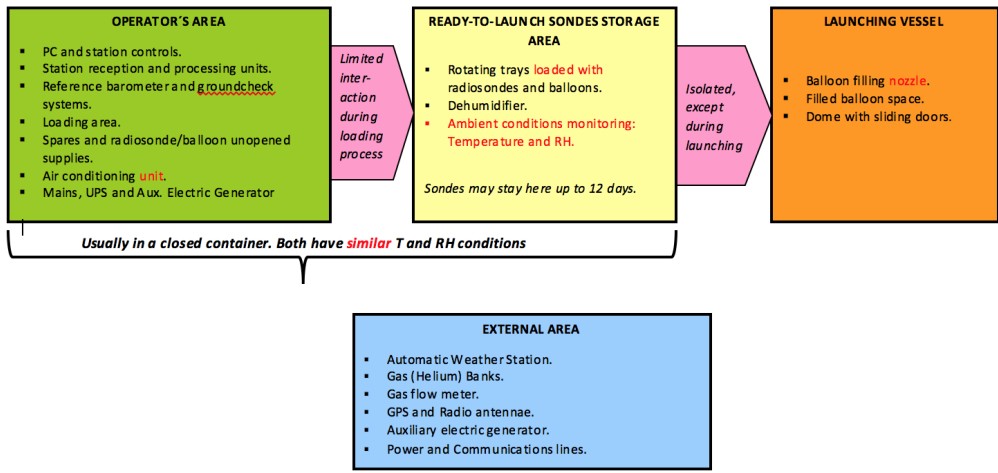

Figure 5: Conceptual diagram of a typical automatic radiosonde launcher divided in four main areas: operator's area
(green), ready-to-launch sondes storage area (yellow), launching vessel area (orange) and external area (cyan).

To assess the effective technical performances of the ARL launches vs manual launches, in Table 4
and 5, examples of the statistics collected at two GRUAN sites running an ARL, Payerne (WIGOS
station identifier=0-20000-0-06610, 46.82N, 6.93E, 490m asl), operated by MeteoSwiss, and
Trappes, operated by Meteo France, respectively, are reported. The Table provides a summary of
pertinent characteristics of the ARL versus manual launches. For Payerne, statistics are related only
to the automatic and manual launches performed since April 2018 (on average, ARL nine per week,
manual five per week) using the Vaisala AS15 ARL. For Trappes, manual launches were performed
in the period 2012-2014, while the Meteomodem Robotsonde has been operated in the period
2016-2018; in both cases two launches per day were performed with similar daily scheduling.
At Payerne, since April 2018 the Vaisala ARL has realized 470 successful flights per year, while
manual launches have been 260 per year. Effective flights according to MeteoSwiss standards are
launches with a balloon burst higher than 100 hPa with no telemetry lost or sensor failure. Despite
the use of different balloon sizes due to the fact that for manual launches bigger balloons are often
used to perform ozonesoundings, the percentage of successful launches as well the percentage of
sondes reaching 10 hPa pressure level is indistinguishable between the ARL and the manual
launches, with a limited use of spare sondes due to the failure of scheduled launches (4 %). Ascent
speed statistics are very close with better performance of the ARL in preventing very low balloon
gas filling.
At Trappes station (Table 5), during the period January 2016 to December 2018, the Meteomodem
ARL Robotsonde in Trappes has realized 1908 successful flights, out of a total of 1956 successful
flights according to MeteoFrance standards (balloon burst at pressure lower than 150 hPa with no
telemetry lost or sensor failure). The mean percentage of successful launches is 97.9% (2016: 95.5%,
2017: 98.2%, 2018: 99.1%, 2019(Jan-Oct): 98.6%, see Figure 6) with an evident improvement using
ARL in the percentage of sondes reaching 10 hPa pressure level (80%) compared to the manual
launches (60 %). The use of Totex balloons is one of the reasons for the improvement and further
improvement was achieved by increasing the size of the balloon. Moreover, since November 2016
Meteomodem has installed a flexible cover which assures that during the storage the balloon is less
exposed to contact with the air-conditioned environment. This seems to reduce the effects of drier
air on the balloon and improve its performance in terms of burst altitude (standard deviation of
burst altitude is reduced after the installation of the cover – not shown). For the balloon ascent
speed, comparison statistics between ARL and manual launches show also similar results.
According to the information shared by Meteomodem, it is also possible to add that, compared to
all the ARLs operated at other sites during the same period reported in Table 5, the Trappes ARL has
typically the same failure statistics. The time evolution of the failure (Figure 6) shows that the
number of spares and the number of failures by type halved in three years to reach less than 2%
relative to the number of successful flights. For the 578 flights performed during 2018, the absolute
number of failures is 2 to the ARL (which was a radio loss and an inflation problem), 1 failure due to
sensor break, no failure from the software, 1 failure which is not classified by our automated failure



identification and 1 failure due to the use of ARL which can be an operator stop or an obstructed
inflation tube.


Table 4: Technical performance of automatic vs manual launches performed at Payerne station
during 2018 for a Vaisala AS15 ARL. Metadata related to the sonde and balloon types are shown
alongside the percentage of success for the launches performed during the reported period, the
percentage of spare sondes used, the sondes bursting before reaching 10 hPa, and the maximum,
minimum and average ascent speed.


| Station | Automatic | Manual |
|---|---|---|
| Station type | AS15 | MW41 |
| RS type | RS41 | RS41 (+ ECC ozonesonde) |
| Balloon type | Totex | Totex |
| Balloon size | 800g | 800g/1200g/2000g/3000g |
| Number of launches | 470/year | 260/year |
| Percentage of successful flights | >99% | >99% |
| Percentage of spare | 4%(spare if P>100hPa) | N/A |
| Sondes above 10 hPa | 92% (based on 2018) | 92% (based on 2018) |
| Max. Ascent speed | 6.1 m/s | 6 m/s |
| Min. Ascent speed | 3.5 m/s | 3 m/s |
| Avg. Ascent speed | 5.2m/s | 5m/s |




Table 5: Same as Table 4 for Trappes site in the period 2016-2018 and 2012-2014, respectively for a
Meteomodem ARL.

| Station | Automatic | Manual |
|---|---|---|
| Station type | Robotsonde (14/04/2015 to 12/2018) | SR10 (01/01/2012 to 14/04/2015) |
| RStype | M10 | M10 |
| Balloon type | Totex | Hwoyee |
| Balloon size | 350g/1000g | Hwoyee 600g |
| Number of launches | 2106 | 2113 |
| Percentage of successful flights | 99% (based on 2018) | >99% (based on 2012) |
| Percentage of spare | 5% (based on 2018) | N/A |
| Sondes above 10 hPa | 80% | 60% |
| Max. Ascent speed | 6 m/s | 6 m/s |
| Min. Ascent speed | 4 m/s | 4 m/s |
| Avg. Ascent speed | 5 m/s | 5.4 m/s |


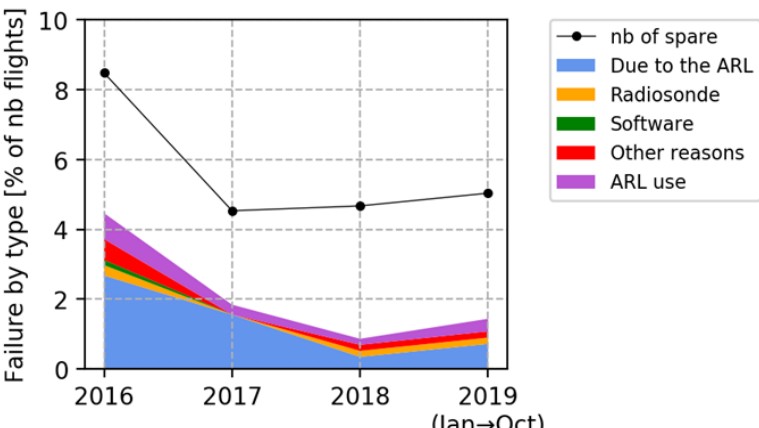


Figure 6: Cause of failure for the Meteomodem ARL in Trappes as a function of time since the
installation date.




## 4. Stability, ground calibration

### 4.1. Performance of the Vaisala ARL

The performance of the Vaisala ARL has been evaluated through the analysis of a dataset collected at Sodankylä station. The Sodankylä Vaisala ARL was used to regularly launch RS92 radiosondes at 11:30 and 23:30 UTC over 2006 to 2012. Manual soundings were periodically performed in parallel using a similar Vaisala DigiCora-3 sounding system throughout this period. Parallel soundings have been selected with launch time difference between 2 minutes and 20 minutes. A total of 283 parallel soundings has been considered: these are distributed evenly across the period, with the exception of 2006, which has more parallel soundings than other years, and most of these are daytime comparisons. In addition, two Vaisala ARL datasets from the Potenza GRUAN station (40.60N, 15.72E, 760 m a.s.l.) and the Minamidaitojima station, run by JMA (WIGOS station identifier index=0-20000-0-47945, 25.79N, 131.22E, 15 m a.s.l.), covering a similar time period, though much smaller sample sizes than in Sodankylä, have been used for comparison. Despite the less intensive sampling, Potenza and Minamidaitojima data are useful data sources to compare with Sodankylä and, specifically, to check consistency of the GC correction across different stations and different batches of Vaisala sondes.

The availability of long time series of parallel sounding for the Sodankylä station permits investigation of the system performance also in the pre-launch phase. Two main aspects are evaluated: stability of the ground check correction on temperature, and potential effects related to the time periods the sondes were stored before launch.

Figure 7 summarises the temperature correction applied during the GC procedure for the RS92 sondes of the above described data sets using the Vaisala GC25 ground check device, with most of the launches performed since 2006. Figure 7 shows similar GC values at Sodankylä, Potenza and Minamidaitojima stations despite the very different locations and launch scheduling, with a negative adjustment of between smaller than -0.5 K before 2010 and smaller than -0.3 K typically applied to most of the RS92 sondes with an improvement of the differences over the time in the batches launched after 2009. The results shown in Figure 7 assume that all the reported ARL GC temperature sensors were maintained according to recommendations described in the previous section.




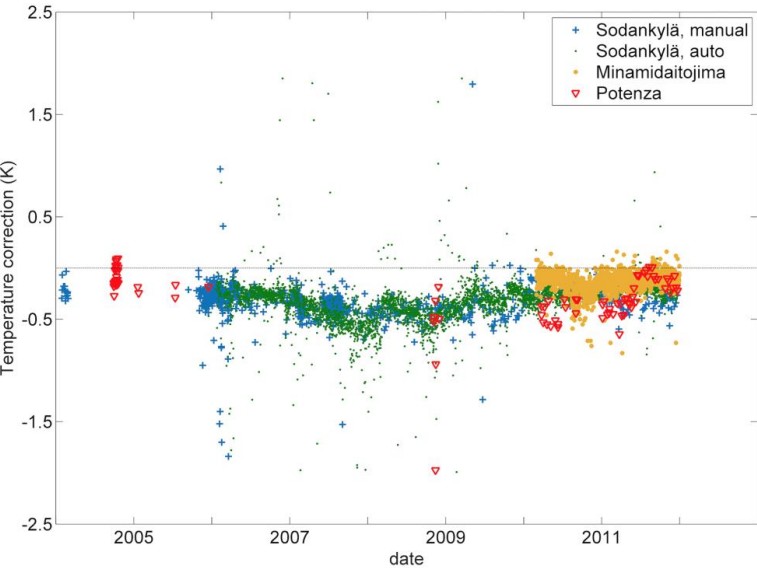


Figure 7: Time series of the temperature correction (temperature measured by the GC reference sensor minus
temperature measured by the sonde) applied during the GC procedure for the RS92 sondes launched at Sodankylä, both
manually (blue crosses) and automatically (green dots), and at Minamidaitojima (yellow dots) and Potenza (red
triangles, automatically) from 2004 to 2012.

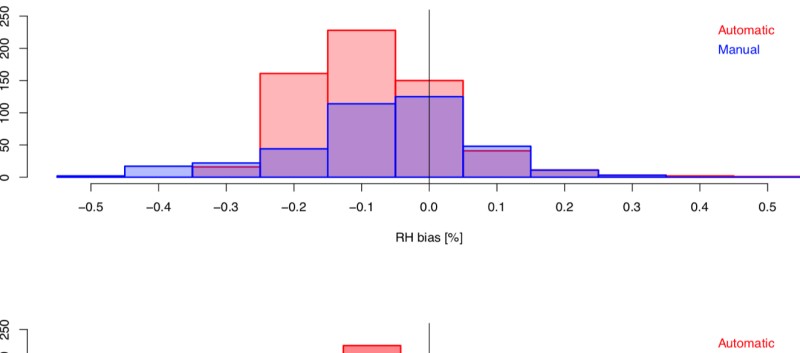


Figure 8: Distribution of temperature and relative humidity corrections found during Vaisala GC process for the
automatic and the manual soundings operated at Payerne station using the RS41 radiosonde.


Results similar to those from Sodankylä and Potenza GRUAN stations are reported by Payerne
GRUAN station (Figure 8) using the RS41 since April 2018 and operating the Vaisala AS15 ARL. Figure
8 shows that the distribution of temperature and relative humidity corrections have negative
skewness with the GC adjustments within a few tenths of a degree and the average adjustment is
smaller than 0.1 K and 0.1% RH, respectively. These results show an average negative GC corrections
for the ARL in analogy to the results reported above for RS92 sondes at Sodankylä and Potenza,
where also the old Vaisala ARL version was operated. Comparisons with the broader statistics
collected by for GRUAN station launching manually (not shown) reveal results consistent with the
GC time series shown in Figure 7 and 8, thus excluding the presence of clear systematic effects in
the GC corrections due to the use of ARLs. Nevertheless, the small differences observed between
the ARL and manual GC corrections needs further investigations to understand if performing the GC
in a controlled temperature and humidity environment may generally improve or worsen the
calibration in the long term.
In an operational station like Sodankylä, the time between balloon loading and ground check can
vary from day to day. At Sodankylä average loading time was 2-3 days prior to launch for regular
soundings. The ARL software allows also longer times in the tray. Figure 9 shows the mean
differences of simultaneous RH profiles (left panel) measured using the ARL and the manual
soundings as a function of the number of days a sonde stays on a tray before launch, from 1 to more
than 5 days. The corresponding standard deviations are also shown (right panel), while in brackets
within the color legend, the number of parallel soundings for each time period is reported. To
calculate the statistics shown in section 4 and 5, radiosounding temperature and RH from parallel
soundings have been interpolated to a 100-meter vertical grid. Figure 9 shows that there are no RH
systematic differences when parallel launches are grouped according to the tray time, except for
the launches with a tray time of 5 days or more at altitude levels above 7 km a.g.l., where a mean
difference smaller than -2.0 % RH is obtained up to 10-12 km a.g.l. Nevertheless, it must be noted
that the size of the sample investigated for these tray time options (5 days and >5 days) is much
smaller than for other tray times and these launches include also parallel sounding with longer
differences in the respective balloon release time. A Wilcoxon Rank Sum Test has been applied and
the computed probability ranges within 0.4-0.5 with smaller values only for above 12 km a.g.l, where
the probability becomes larger than 0.2. For the time tray option with a smaller sample of parallel
soundings (1 day, 5 day and >5 days), the probability oscillates between 0.05 and 0.10. Therefore,
it is possible to conclude that we do not reject the hypothesis that the two data distributions (ARL



and manual launches) have the same median value and the reported comparisons are meaningful.
Finally, the right panel of Figure 4 show that the standard deviations are substantially smaller than
5% RH at all altitude levels without any evident correlation with tray time. A similar test for longer
storage time, up to one month, has been carried out recently in Sodankylä providing similar GC
results (not shown).

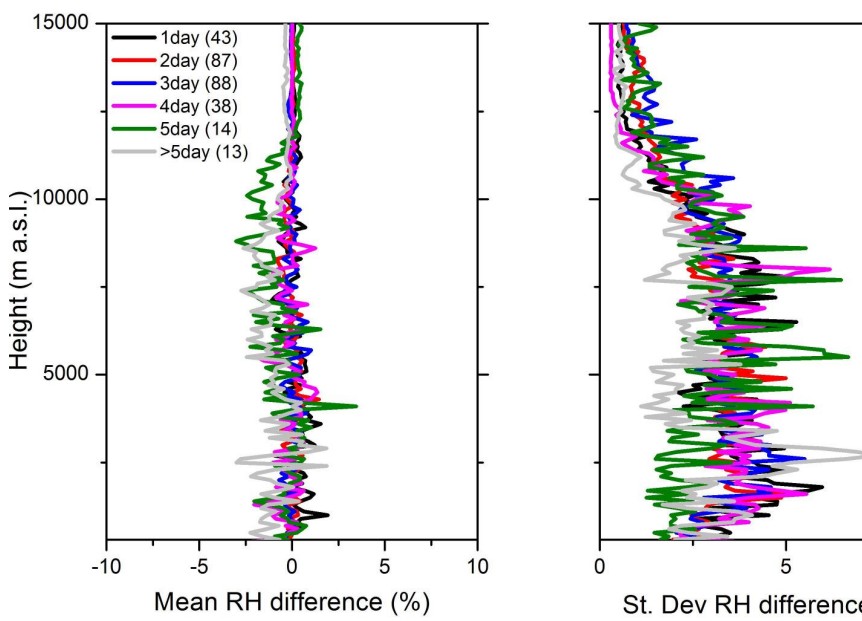

Figure 9: Vertical profiles of the mean difference and standard deviation of the RH measured with the manual and
automatic system in Sodankylä as a function of the time period between GC and launch; from left to the right, the time
period increases from 1 to more than 5 days. In brackets within the legend, the number of parallel soundings considered
for each time period is reported.

In Figure 10, a similar study to that reported in Figure 9 is presented for the Payerne station. In this
case, the average difference and the standard deviation of temperature and relative humidity found
during the GC using Vaisala RS41 radiosondes into the Vaisala AS15 versus the aging (up to 9 days
into tray from the loading until launch) is shown. For both temperature and relative humidity,
excluding only the launches which occurred within 24 hours of the radiosonde loading, the bias is
negative and independent of any further aging. Until one day after loading the bias is stable close
to zero and thereafter it increases to about -0.1 K and -0.1% over the following days. These results
show how the use of ARLs also in remote places or where it is required to upload in advance a large
number of radiosondes, to launch with a few days of delay, do not appreciably lead to changes in
the Vaisala GC.



In Figure 10, a similar study to that reported in Figure 9 is presented for the Payerne station. In this
case, the average difference and the standard deviation of temperature and relative humidity found
during the GC using Vaisala RS41 radiosondes into the Vaisala AS15 versus the aging (up to 9 days
into tray from the loading until launch) is shown. For both temperature and relative humidity,
excluding only the launches which occurred within 24 hours of the radiosonde loading, the bias is
negative and independent of any further aging. Until one day after loading the bias is stable close
to zero and thereafter it increases to about -0.1 K and -0.1% over the following days. These results
show how the use of ARLs also in remote places or where it is required to upload in advance a large
number of radiosondes, to launch with a few days of delay, do not appreciably lead to changes in
the Vaisala GC.

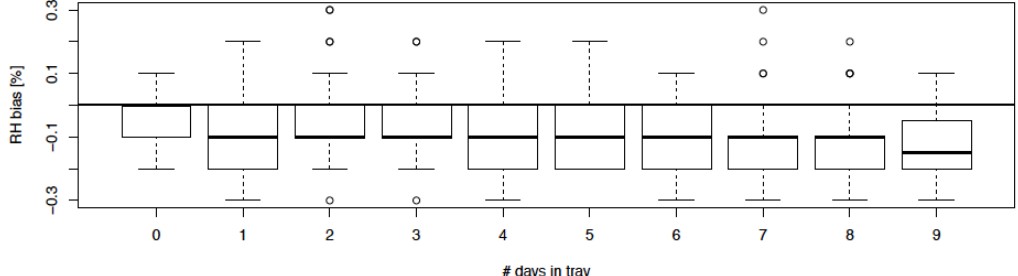

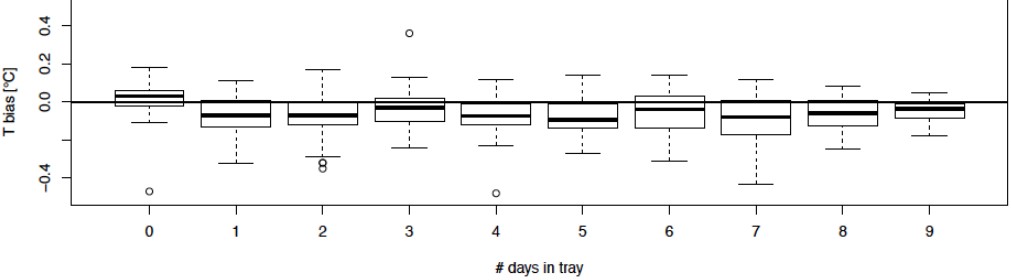

Figure 10: Average difference and standard deviation of temperature and relative humidity found during the Vaisala GC
process versus the aging (number of days into tray from the loading until launch) of the radiosonde RS41 into the
Payerne ARL (Vaisala AS15).

### 4.2.    Performance of the Meteomodem ARL

The performance of the Meteomodem ARL ground-check has been evaluated through the analysis
of a dataset collected at MeteoFrance Trappes station, where M10 radiosondes have been launched
regularly at 11:30 and 23:30 UTC since 2016. The availability of a long time series for the comparison



between M10 temperature and humidity sensor and one reference temperature/humidity sensor
(Vaisala HMP110, https://www.vaisala.com/sites/default/files/documents/HMP110-Datasheet-
B210852EN_1.pdf) at ambient conditions, inside a meteorological shelter for the Trappes station,
permits the investigation of the system performance also in the pre-launch phase. Since June 2018,
this comparison is carried out during the 5 minutes before each automatic sounding. Figure 11
summarizes the time series and PDF of the difference between M10 and HMP110 sensor for
temperature (black curve, upper panel) and relative humidity (blue curve, lower panel) recorded
between June 2018 and June 2019. The relative humidity difference oscillates around 0% and in
more than 75% of the cases the difference is smaller than 2% RH in absolute value. For temperature,
the observed residual difference around 0.5°C requires further investigations.

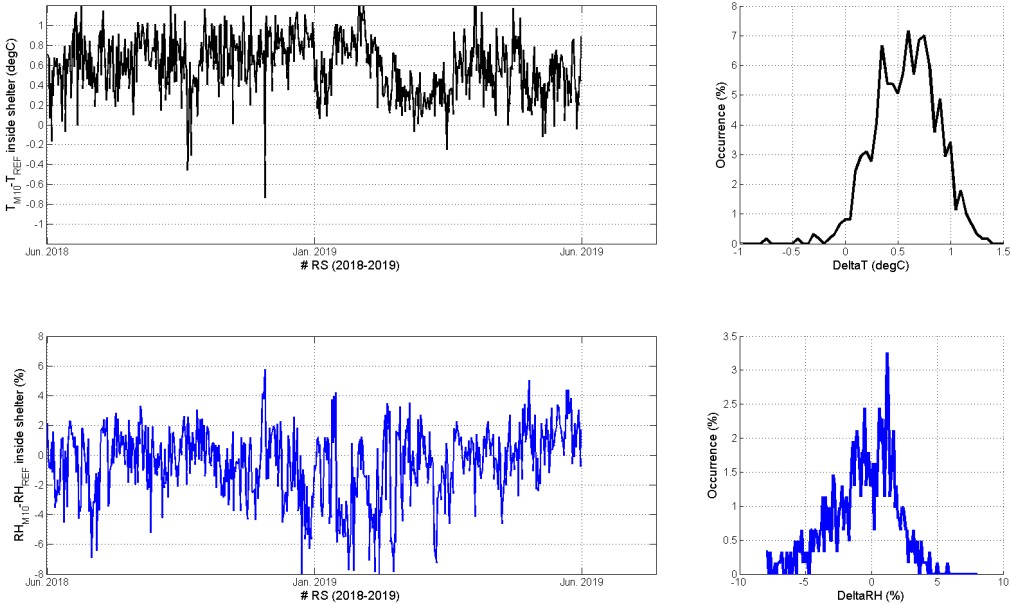


Figure 11: Time series and pdf of the difference between M10 and HMP110 sensor for temperature (black curve) and
relative humidity (blue curve) between June 2018 and June 2019, measured at ground level inside a meteorological
shelter in ambient condition.

Figure 12 provides a picture of the meteorological shelter and the position of the HMP110 and the
M10 during the 5-minutes comparison shown in Figure 11. These results need further investigations
in order to determine if the systematic difference observed on temperature in the meteorological
shelter is due to the Meteomodem M10 batches produced in 2018, though Meteomodem did not
report similar systematic difference during the production checks, or if this could be due to the need





of improving in the experimental protocol. The meteorological shelter has been improved with the
installation of a fan (Figure 12) which should produce a better homogenisation of the temperature
and relative humidity around the two sensors. The development of a new experimental protocol is
under consideration and should lead to the production of a tube ventilated by a laminar flow in
which the Meteomodem M10 and a PTU reference could measure under the same environment,
upon the characterization of the spatial homogeneity of the temperature and relative humidity.

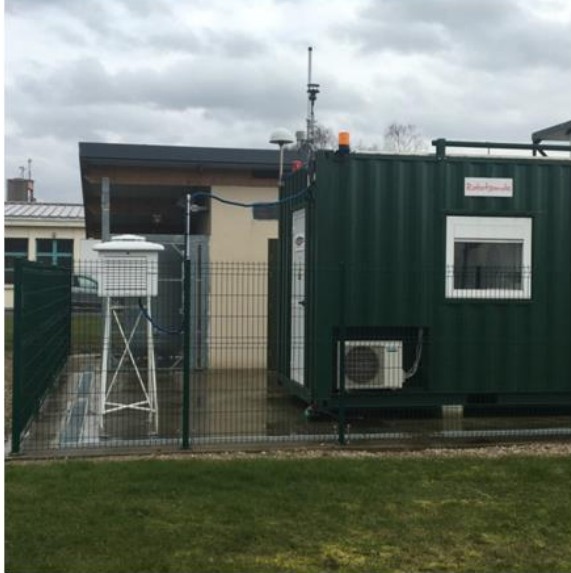
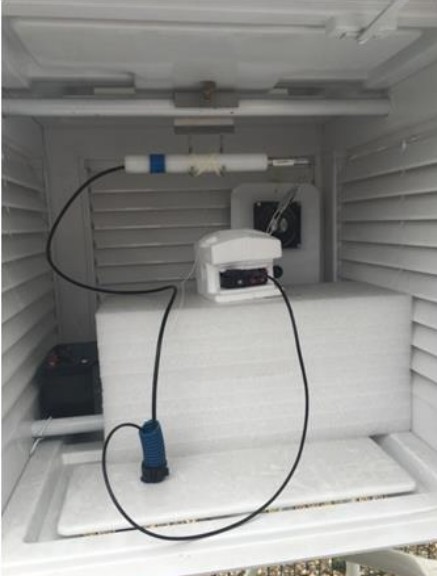


Figure 12: Picture of the meteorological shelter in Trappes (left panel: general view: the meteorological is near the
Meteomodem ARL entrance for simplicity reasons, right panel: inside of the meteorological shelter)

Finally, the M10 radiosonde is put inside a SHC chamber for 3 minutes before the sounding (with a
relative humidity near 100%): more than 95% of the samplings are accepted after the test. For
operational reasons, the Meteomodem probes used in the GRUAN protocol, are tested in the
meteorological shelter and in the 100% RH test but not necessarily in this order at each time. It is
not known if the order of the checks makes any difference.

**5.    Vertical velocity and balloon burst**
This section reports the statistics for the vertical velocity and the balloon burst altitudes from the
datasets collected at Sodankylä and Trappes stations.



**5.1 Vertical velocity and balloon burst altitude for Vaisala technology**
In Figure 13, the statistics of the balloon vertical velocity and of the burst altitude for Sodankylä in
the period from 2006 to 2012 are shown. In terms of vertical velocity (Figure 13, left panel), the ARL
has a quasi-symmetric frequency distribution peaked around 5.3 m s$^{-1}$ with a spread mainly between
4.7 m s$^{-1}$ and 5.9 m s$^{-1}$. For the manual launches, the frequency distribution is quite wide, non-
symmetric, peaked around 4.5 m s$^{-1}$ with a larger spread of the values mainly between 3.5 m s$^{-1}$ and
5.7 m s$^{-1}$. The comparison reveals the higher stability of the ARL compared to manual launches in
controlling the balloon filling and, therefore, the sounding vertical velocity which is relevant for the
quality of the measured profile. For the balloon burst altitude (Figure 13, right panel), a real
comparison between the manual launches and the ARL is not feasible at Sodankylä due to the use
of different balloon types (typically smaller for the ARL) which causes a strong difference in balloon
altitude. Totex Tx800 or Tx600 type of balloons were used in winter and Totex Ta350 or Tx350 type
sounding balloons were flown during all other seasons. Due to smaller balloon volume, the
summertime soundings had lower burst heights on average. The burst altitude for the ARL has also
in this case a quasi-symmetric frequency distribution peaked around 25 km of altitude a.g.l with a
spread of the values mainly between 17 km and 28 km a.g.l., while the distribution for manual
launches is non-symmetric, with a maximum frequency around 33 km and most of values ranging
within 21 - 35 km a.g.l. It must be mentioned that the differences between night-time and day-time
soundings were not significant, although night time soundings have on average lower burst heights
during polar vortex overhead conditions in winter.

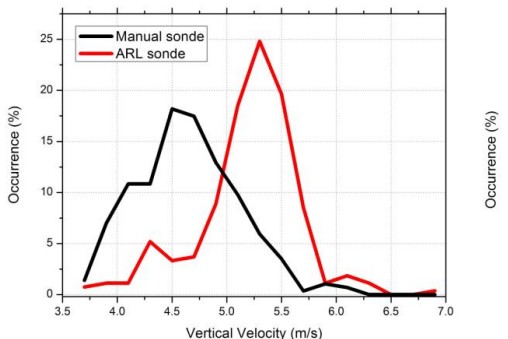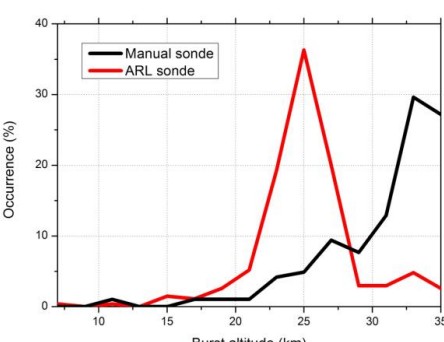


Figure 13: Vertical velocity (left panel) for radiosondes launched manually (black line) and automatically (red line), along
with burst altitude (right panel) at Sodankylä station.



### 5.2 Vertical velocity and balloon burst altitude for Meteomodem technology

A more interesting comparison to show the positive influence of automation on the burst altitude is those related to the dataset discussed in Section 3 and summarized in Table 5, shared by Meteo France for Trappes station (Figure 14). In terms of vertical velocity (Figure 14, left panel), both the ARL and the manual launches have a quasi-symmetric frequency distribution peaked around 5.5 m s$^{-1}$ and 5.1 m s$^{-1}$, respectively, with a similar spread of about 1.0 m s$^{-1}$. For the burst altitude (Figure 14, right panel), we have for both the dataset a negatively skewed distribution with an evident peak around 33 km for the manual launches and 35 km for the ARL. The comparison reveals that the burst altitude (Figure 14, right panel) is significantly higher and less scattered after the automation, while the vertical velocity of the balloon has not significantly changed (Figure 14, left panel). 40 % of the balloons burst before 30 km during the manual period, where only 20 % during the automatic period, this result means that the Meteomodem ARL and/or the operational organization has increased by a factor two the number of balloons reaching an altitude higher than 30 km. The burst altitude for both periods (2012-2014 for the manual launches and 2016-2018 for the ARL) shows some seasonal signal. It appears that burst altitude is lower during the winter. A further study could evaluate burst altitude as a function of air temperature or potential vorticity in order to study the influence of polar vortex and its potential impact on the burst altitude.

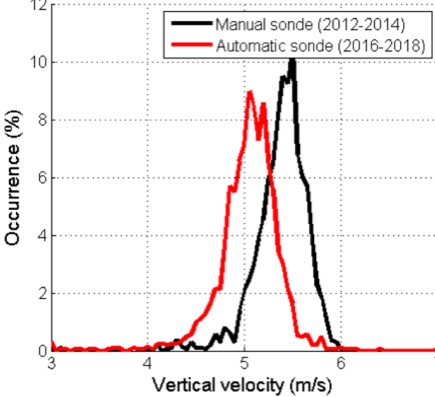
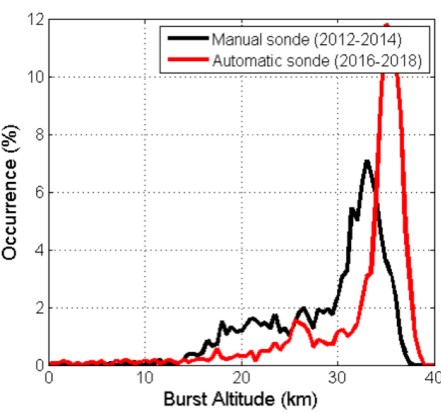

Figure 14: Vertical velocity (left panel) for radiosondes launched manually (black line) and automatically (red line), along with burst altitude (right panel) at Trappes station.



### 5.3 Quantifying relative performance

In this section, two datasets are investigated to assess the differences in the vertical profiles of temperature and humidity: the set of RS-92 parallel (automatic and manual) soundings performed with the automatic radiosonde launchers at Sodankylä along with a second set of Meteomodem radiosoundings collected at Faa'a station, French Polynesia. In the following analysis, given the latitude φ, the longitude λ, the Earth's radius R (mean radius = 6371 km), the distance between two balloons (1 and 2) has been calculated using the 'haversine' formula (Sheppard and Soule, 1922) which provides the great-circle distance between two points (i.e., shortest distance over the earth's surface):

$$d = Rc$$

where

$$c = 2 atan2(\sqrt{a}, \sqrt{(1-a)})$$

$$a = sin^2\left(\frac{\Delta\lambda}{2}\right) + cos(\varphi 1)\, cos(\varphi 2)\, sin^2\left(\frac{\Delta\lambda}{2}\right)$$

The haversine formula remains particularly well-conditioned for numerical computation even at small distances – unlike calculations based on the spherical law of cosines. The function "atan2" is described in Glisson (2011).

The two datasets are also investigated to show the correlation between the difference in the vertical profiles and the distance between the two flying sondes.

### 5.1 Parallel soundings with Vaisala systems

For the same six-year dataset collected at Sodankylä discussed in Section 4, the vertical profiles of the average differences (automatic minus manual) and standard deviations of the temperature and RH measured during parallel soundings are shown in the left panel of Figure 15. Systematic differences in the temperature profile are negligible (on average smaller than 0.01 K) over the entire vertical range up to 25 km a.g.l, while the standard deviation increases with altitude from values smaller than ±0.5 K below 15 km to values larger than 1 K above. The result is in agreement with the increase in mean distance between near simultaneous sonde paths at higher altitudes (Figure 16). A subset of the parallel temperature soundings at Sodankylä has previously been analyzed by Sofieva et al. (2008). Even though it is hard to separate components from non-colocation from those which may arise from instrument-to-instrument differences (e.g. arising from manufacture variations and differences in preparation, storage and launch at the uppermost altitudes), Sofieva



et al. found differences in small scale structures in temperature profiles, when the horizontal
separation was larger than 20 km. Moreover, to investigate whether the ARL and the manual
radiosoundings datasets were selected from populations having the same distribution, i.e. if the
calculated mean differences are statistically significant, the Wilcoxon Rank Sum test has been
applied: this confirm that the two datasets are samples of the same population showing a
probability larger than 0.5 for temperature at all the altitude levels below 20 km and values larger
than 0.1 above, while for RH values larger than 0.3 over the entire range from surface to 15 km a.g.l

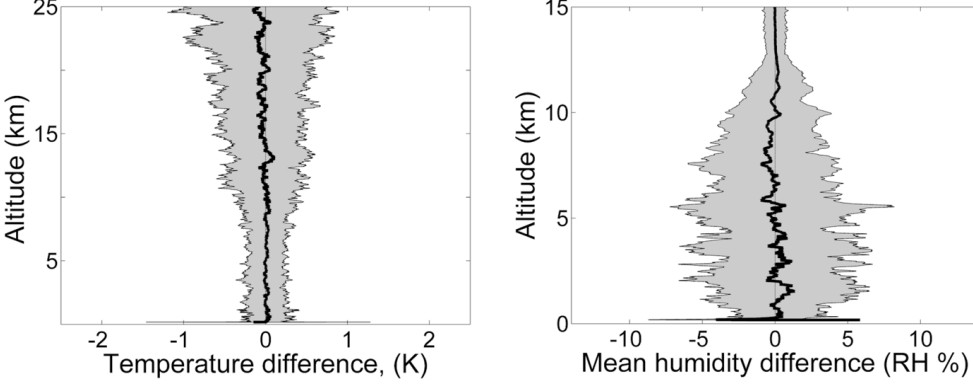


Figure 15: Temperature (left panel) and RH (right panel) mean difference between ARL and manual for the six-year
dataset of parallel soundings collected at Sodankylä station at all altitude levels up to 25 km a.g.l for temperature and
up to 15 km a.g.l for RH.  Standard deviation at each pressure level is reported using the gray area.

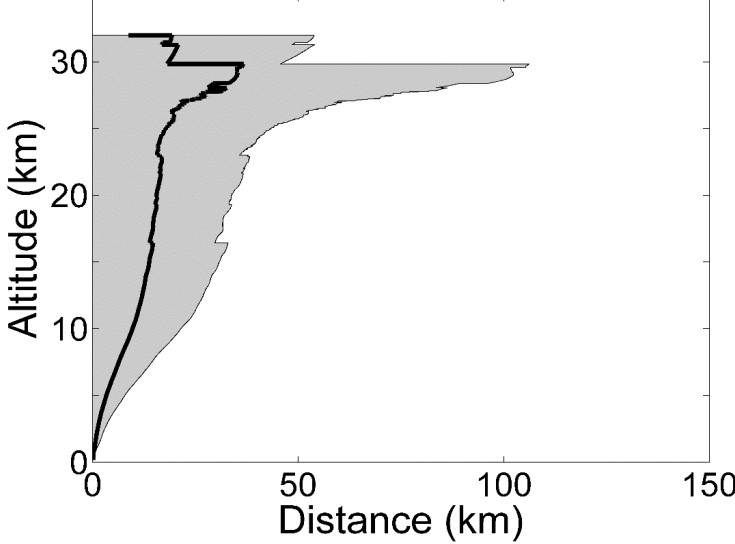


Figure 16: Horizontal distance between the balloons calculated for the six-year dataset of parallel soundings collected
at Sodankylä station for all the altitude levels up to 32 km a.g.l.



For the RH mean difference profile (Figure 15, right panel), there are no significant systematic
differences up to 7 km and then again above 10 km a.g.l., while in between these altitudes a small
negative mean difference lower than 1% RH is found and may be related to the coupling between
the RH variability in the upper troposphere and the distance between the two sondes. The increase
in standard deviation the lower troposphere below 5 km a.g.l., with values generally smaller than
5% RH, it is due to the high RH variability which can be significant even for small horizontal distances
between the two sondes. Above 5 km, going to the UT/LS where the values of RH are on average
smaller and less variable, RH difference decreases except when clouds or other uncommon events
are detected (e.g. Stratospheric-Tropospheric exchanges).
In addition, the analysis was rerun after grouping the ARL flights according to the time a sonde had
been loaded to the launcher system (see section 4): variations of time period between sonde loading
and actual launch time did not influence the comparison results.
Finally, the Wilcoxon Rank Sum Test has been applied to the entire dataset and the computed
probability that the two samples belong to the same population is larger than 0.35 at all altitude
levels.

**5.4 Parallel soundings at Faa'a with Meteomodem systems**
A first evaluation of the performance of Meteomodem ARL is provided by the analysis of the
datasets collected over 3-14 October 2018 at Faa'a station (French Polynesia, 28.34S, 16.32E, 21 m
a.s.l.) where 21 launches (9 day-time and 12 night-time) of parallel radiosoundings have been
undertaken (a picture is provided in Figure 17) in order both to compare temperature, relative
humidity, wind speed and direction, and to study further characteristics of the flights (burst altitude,
ascent speed for example). Meteo-France has conducted the Intensive Operational Period while
Institut Pierre Simon Laplace (IPSL) has produced the NetCDF files (data and metadata) for the
analysis. Raw data without any correction for temperature and relative humidity have been
considered in this paper. The GRUAN data processing, which remains under development at the
present time for this datastream, has not been applied. The manufacturer Meteomodem IR2010
software was used for both manual and automatic launches.
ECMWF noted that some reports from Meteomodem Robotsondes at other stations had
anomalously dry, and sometimes warm, values just above the surface relative to the background
field. In cool, moist atmospheric conditions the anomalies can be two or three degrees for
temperature and larger for dew point temperature. "For technical reasons the launcher has to be
kept warm and dry internally, which means that the humidity sensor is initially reading quite low





and a bubble of warm/dry air escapes with the balloon at launch - the net effect is that the first few
decametres the dewpoint reading is too low." (Ray McGrath, pers. comm. 2015). The issue
described above does not affect the proflle at higher levels. A similar issue has also been reported
for data taken during the first few seconds with Meisei ARL and this is suspected to be due again to
the influence of the air inside the launcher.

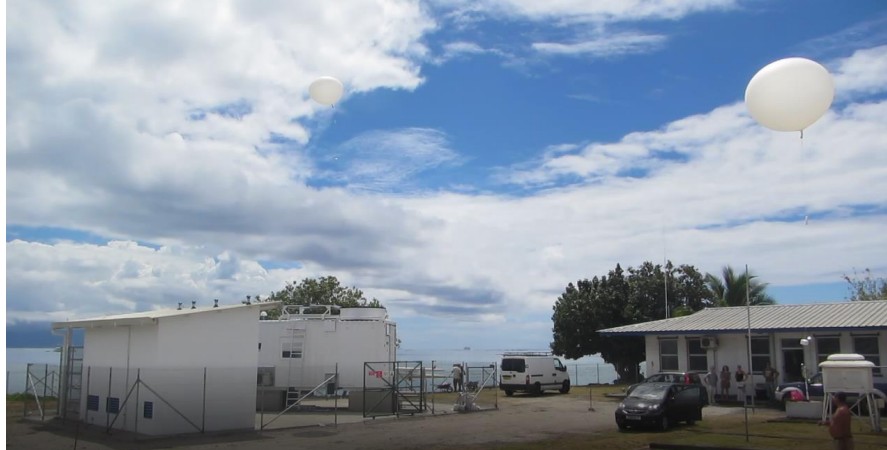


Figure 17: Daytime parallel sounding at Faa'a station (French Polynesia).

The Meteomodem has recently implemented a new software, EOSCAN, not yet implemented at all
the stations, which improves the ARL dataset quality with a number of corrections such as:
1. Eliminating the GPS disturbances at the end of the tube that can persist in the first 20 seconds
after the release;
2. Adjusting for the systematic bias introduced by the fact that the ARL Meteomodem is air
conditioned and affecting the first 150 m of the radiosounding profiles.
The dataset collected by Meteo-France at Faa'a station is not sufficiently large to draw robust
statistical inferences. Nevertheless, this dataset is the first ever available to evaluate the
performances of the Meteomodem ARL and can provide useful indications of any likely impact upon
the data quality of ARL facilities.
Before comparing, the T and RH profiles of the parallel sounding dataset have been interpolated to
a resolution of 100 m altitude. The difference between the launch time of the ARL and the manual
balloons ranges within 1 and 12 seconds.




In Figure 18, the horizontal distance between the pairs of parallel soundings at all the altitude level
up to 25 km a.g.l is shown: the horizontal distance between the two balloons is typically within
about 35 km.
In Figure 19, the difference between the balloon automatically released and the manual one as a
function of altitude regardless of time mismatch, for each pair of parallel soundings is reported
(black line) along with the corresponding mean difference (grey dashed line); the left panel shows
the difference for temperature, while the right panel for RH. The mean temperature difference is
smaller than ±0.2 K up to 12-13 km a.g.l., and typically smaller than ±0.5 K above. The difference is
negative, up to -2.0 K, in the first 50-100 meters and this is probably due to the potential warming
effect of the ARL environment on the radiosonde sensor.

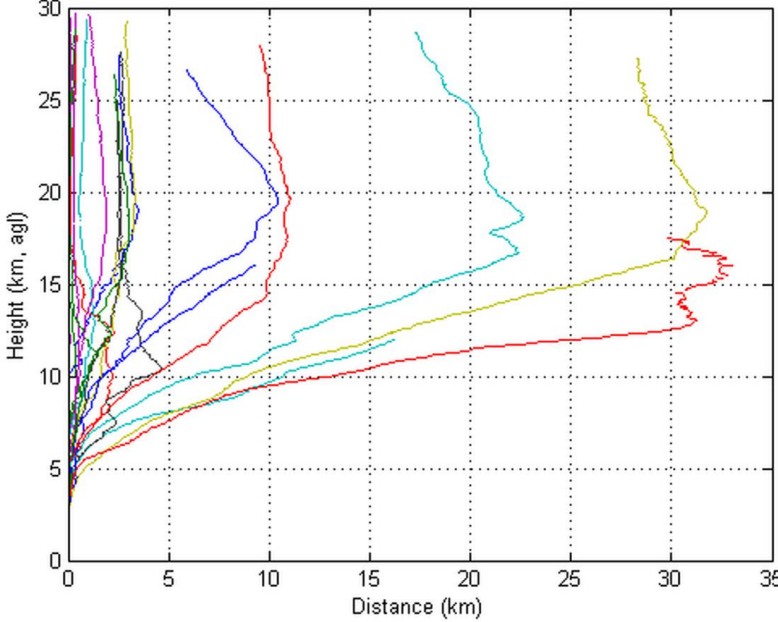


Figure 18: Horizontal distance calculated for the balloons of the 21 parallel soundings performed at Faa'a station for all
the altitude levels up to 25 km a.g.l. Measurement time between the two sondes at the same altitude levels may differ
and at the start time ranges within 1-12 seconds.

For RH, the mean difference is instead always positive and smaller than 0.7% RH up to 8 km a.g.l.
with a standard deviation smaller than 3-4% RH. Above 8 km, the mean difference becomes larger
and less variable with a maximum of about 2% RH and a standard deviation around 3%. The
Wilcoxon Rank rank sum test has been applied to both temperature and RH. For temperature, the



probability is higher than 0.3 until 17 km and higher than 0.2 above, while for RH is larger than 0.2
below 10 km and larger than 0.1 above. Only in the first 40 m for temperature and the first 20 m for
RH, the Wilcoxon Rank rank sum test fails with a probability lower than 0.05. The results of the test
allow to confirm the null hypothesis of the same median for the ARL and manual data distribution
at all the height levels for both temperature and RH, with the only exception of a few decameter
above the ground because of the ARL air conditioned effect. The reason behind this bias could arise
from GC effects or differences in the pre-launch procedures between the two systems affecting the
performance of one of the two launches in a quasi-systematic manner throughout the vertical
profile. This will be further investigated with the support of the manufacturer.
In terms of balloon burst altitude the ARL proved to be reliable both during the daytime with a burst
altitude ranging within 26688 - 31904 m above ground level (a.g.l.) versus values within 24970 -
30621 m a.g.l. calculated for the manual launches, while during nighttime the burst altitude ranges
within 27587 - 30790 m a.g.l. for the automatic launcher versus values within 27437 - 30139 m a.g.l.
for the manual launches. Applying the Wilcoxon Rank-Sum Test, the computed probability (0.05224)
is slightly greater than the 0.05 significance level and therefore we do not reject the hypothesis that
the two distributions of burst altitude values, for ARL and manual launches respectively, have the
same median value, indicating that ARL does lead to improvements in the balloon burst altitude.

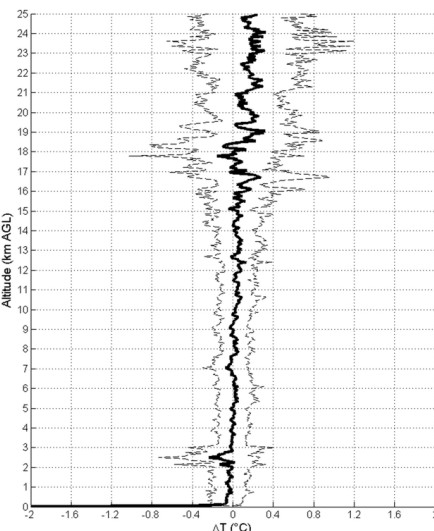 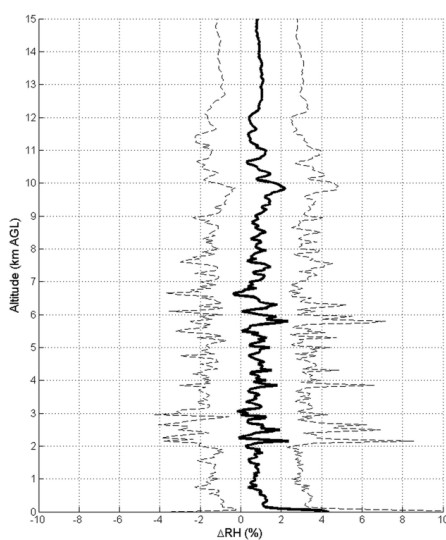

Figure 19: Difference between ARL and manual profiles of temperature (left panel) and RH (right panel) for 21 parallel
soundings performed at Faa'a station up to 25 km a.g.l. for temperature and up to 15 km a.g.l. for relative humidity.
Black lines: mean differences, dashed lines: standard deviation.




## 6. Automatic launchers performance evaluated using the ECMWF forecast model

Data assimilation systems compare observations with a short-range forecast (called the
background) and use observation-minus-background (O-B) differences in the assimilation to provide
improved initial conditions for the next forecast. For some areas/variables the uncertainties in the
background are now similar to, or smaller than, those in the observations, so the background
provides a very useful comparator. O-B differences from reanalyses have been also used to
homogenise historical radiosonde data (Haimberger et al., 2012). Ingleby (2017) compared different
radiosonde types with ECMWF background fields and for temperature and upper-tropospheric
humidity found differences in radiosonde performance that are broadly consistent with the results
of the last WMO radiosonde intercomparison (Nash et al., 2011) and are dominated by the sonde
type.
Statistics for Vaisala and Meteomodem radiosondes (manned and ARL) were produced. For Vaisala
we examined the German radiosondes (Figure 20) which form a relatively dense, well maintained
network with manned and ARL stations interspersed - ideal for this type of comparison. The
background uncertainties vary somewhat over time and regionally - they are probably slightly larger
over the UK because of the proximity of the North Atlantic. The Meteomodem samples were quite
small (from five French stations in total) and inconclusive; therefore, they will not be shown. No
attempts to provide a comparison of O-B statistics for Meisei ARL station were carried out. This is
due to the fact that all four Meisei ARLs are on small islands, three to the south of the main islands
of Japan and one to the south-east, whereas the manned stations are on the main islands (or two
distant islands). Therefore, the O-B comparison could be affected by differences in the
background uncertainties over the southern islands relative to the main islands.
Figure 21 shows the numbers of reports at standard levels for German RS92 launches in the period
2015-2017. There are more than twice as many manned launches as ARL ascents because four of
the manned stations usually report four times per day whereas the other four manned stations and
the five ARL stations report twice a day. One interesting feature is that the proportion of ARL ascents
reaching 20 hPa is significantly higher than the proportion of manned ascents. A plausible
explanation for this is that ARLs put less stress on the neck of the balloon than manual launches
(Tim Oakley, pers. comm. 2018). During the middle months of 2017, there was a transition from
Vaisala RS92 to Vaisala RS41 at German stations - the proportions of RS41 reports at different
standard levels (not shown) are very similar to those in Figure 20.



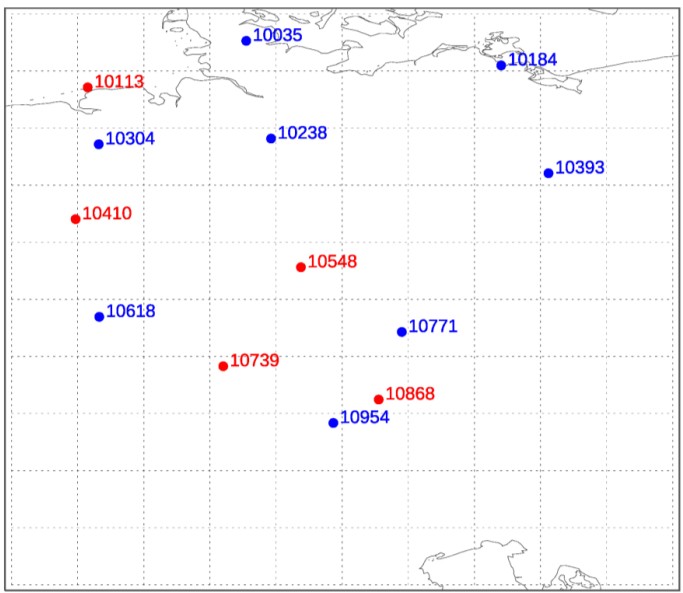


Figure 20: The main German radiosonde sites (two training/test sites not shown) and station identifiers: blue - manned
stations (8), red - autosondes (5), as in early 2019 and for several years before that.

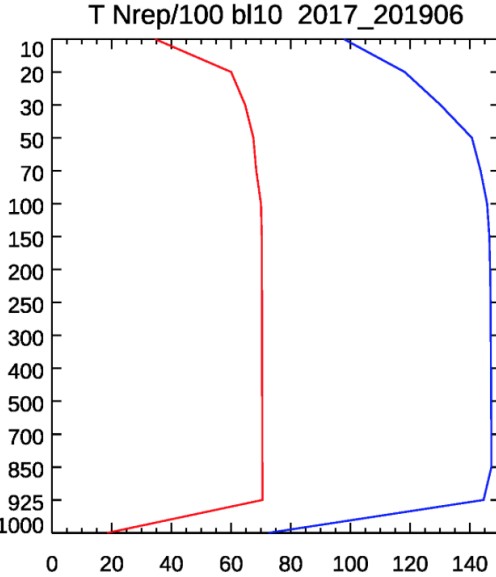


Figure 21.  The number of temperature reports (hundreds) at standard levels, hPa, from German stations using Vaisala
RS92 radiosondes, 2015-2017: blue - manned stations, red - autosondes. The numbers for other variables are very
similar. There are fewer reports at 1000 hPa, and to some extent at 925 hPa, because these levels can be below the
launch site.  The decrease at upper levels is due to balloon burst.





Figures 22 and 23 compare O-B mean and root-mean-square (rms) statistics for German RS92 and
RS41 reports respectively (for technical reasons alphanumeric TEMP reports were used rather than
binary BUFR reports, see Ingleby and Edwards, 2014). The RS92 results (Figure 22) are very similar
between manned and ARL stations (small differences at 1000 hPa are presumably due to the
proximity of the surface and relatively small samples). The upper tropospheric humidity has minor
systematic differences probably due to humidity time-lag and radiation corrections being
introduced at different dates at different stations.

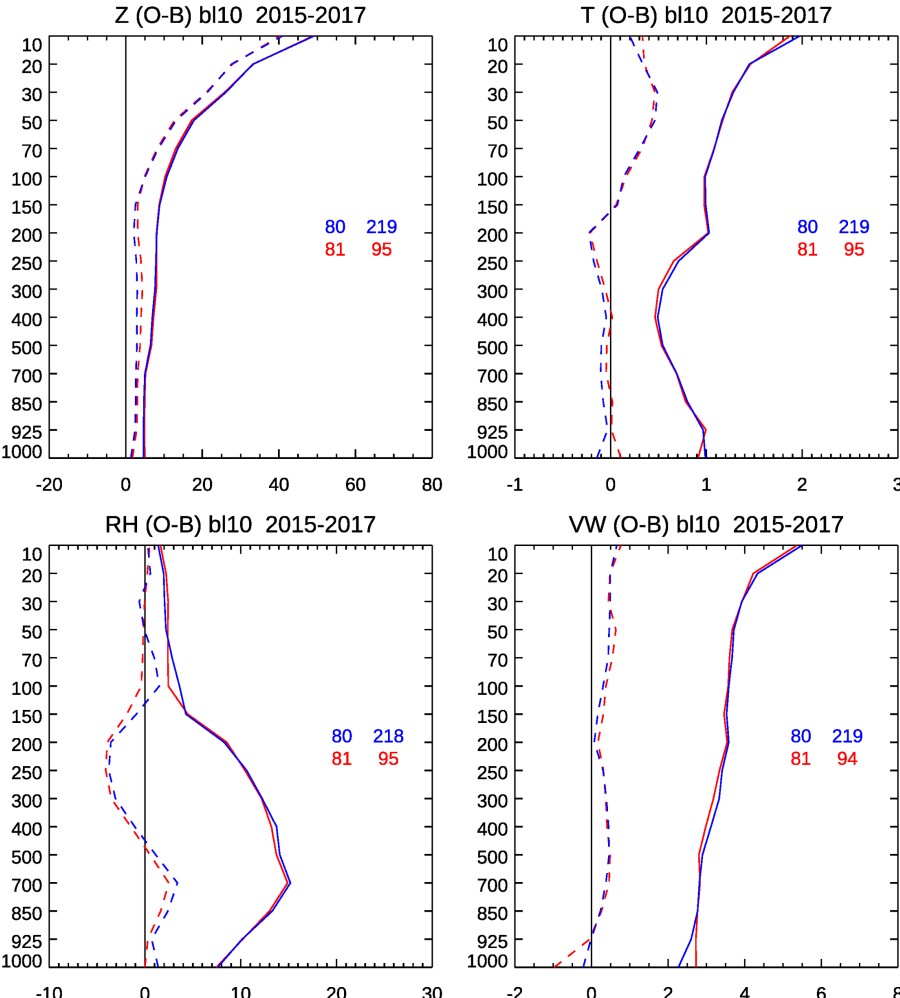


Figure 22: Mean (dashed) and rms (solid) O-B statistics for German RS92 ascents, 2015-2017: blue - manned, red - ARL.
Results for geopotential height (top left), temperature (top right), relative humidity (bottom left) and wind (mean wind
speed and rms vector wind; bottom right). The key gives the radiosonde code (80 for manual or 81 for ARL) and the
number of reports in hundreds.





In contrast and surprisingly, the RS41 results (Figure 23) show rather larger rms(O-B) differences for
ARL stations - especially for temperature and wind. Qualitatively similar results for RS41 are found
for subsets of the period considered confirming the robustness of the results. The reasons for the
larger ARL rms differences in Figure 23 are not clear yet; one possibility is linked to the accuracy of
the reported pressure values. Pressure is measured by the RS92. For the RS41-SG the pressure is
calculated starting from a surface pressure measurement, but the German stations use the RS41-
SGP with a pressure sensor. Discussions with Vaisala and DWD (the German weather service) have
not so far revealed the cause.

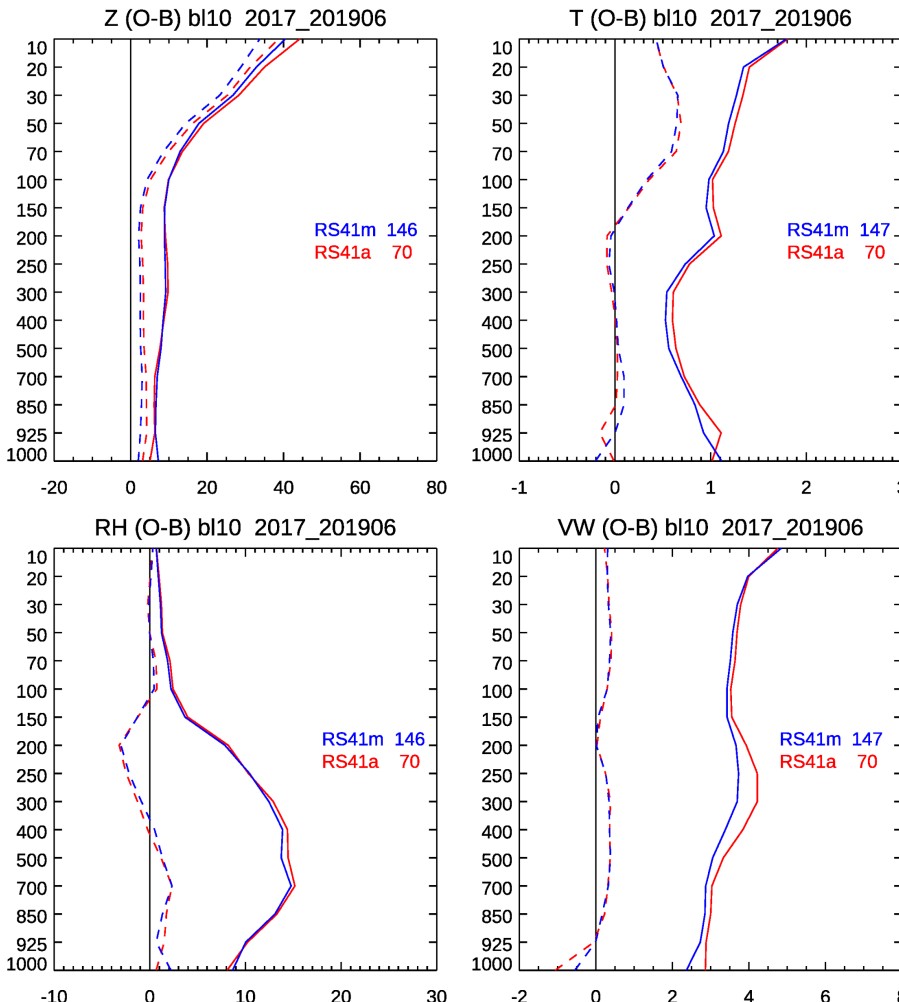

Figure 23: As Figure 22 but for RS41 reports, 2017-June 2019.  For some months, all stations reported as type 23 (123
in BUFR) so they had to be separated using the station identifiers.



## 7. Summary and discussion

In this paper, the existing Automatic Radiosonde Launchers available on the market (Vaisala, Meteomodem and Meisei) are presented and a first comparative analysis of the performance, relative to the more prevalent practice of manual launches, for the two most mature systems at present (Vaisala and Meteomodem) has been reported. The analysis is limited to the data available from a few GRUAN certified or candidate sites (Sondakyla, Payerne, Trappes, Potenza, Faa'a) and to the investigation of the O-B bias and rms using the ECMWF forecast model and the Vaisala ARLs and manual stations of the DWD. The data analysis allows to infer the following principal conclusions:

- From a technical point of view, the performance of ARL is fully similar or superior to that achieved with the traditional manual launches due to the capability of the automatic launchers to fully control several parameters during the different phases of the radiosonde preparation and balloon launch. This reduces launch-to-launch variability typical in manual launches.

- Despite having some potential advantages, there are still some issues generating failure in the launches which can be improved according to the feedback provided by the GRUAN sites, operating mainly Vaisala ARLs, such as the not infrequent failure of the power supply system or of the air conditioning system, plenty of issues related to the balloon release in the vessel area, likely contributing to early balloon bursts, and to the management of the gas flow to fill the balloon, while the ready-to-launch sondes storage area appears to be the most efficient part of ARLs.

- For both temperature and relative humidity, the GC correction has been investigated for the Vaisala ARL, finding a negative offset relative to manual launch procedures at different stations and considering different radiosonde types (RS92/RS41) and batches of a few tenths of degree and % RH, respectively. For the Meteomodem ARL at Trappes station, the difference between M10 temperature and humidity sensor and the Vaisala HMP110 housed in the ARL, used as a reference immediately prior to launch shows a few tenths of degree and % RH, respectively. These results need further investigation to understand the underlying reasons and whether manual or ARL operations are closer to the observed atmospheric profiles.

- Systematic differences in the temperature profile for both Meteomodem and Vaisala are smaller than ±0.2 K up to 10 hPa; RH difference profile differences are smaller than 1% RH for the Sodankylä Vaisala dataset up to 300 hPa, while it is constantly positive and smaller



than 2% for Faa'a station Meteomodem series. However, the restricted dataset available at
Faa'a station means caution should be applied in generalizing these results as representative
of all Meteomodem ARL.
● O-B mean and rms statistics for German RS92 and RS41 are very similar between manned
and ARL stations. The upper tropospheric humidity has minor systematic differences
probably due to humidity time-lag and radiation corrections being introduced at different
dates at different stations. The RS41 sondes shows larger rms(O-B) differences for ARL
stations than RS92, in particular for temperature and wind.  The accuracy of the reported
pressure values might be a possible reason to explain this difference.

As mentioned at the beginning of section 3, the factor limiting adoption of ARL radiosounding
products within the GRUAN reference network is mainly related to the use of an independent and
traceable calibration standards like the Standard Humidity Chamber (SHC) within the ARLs. At
present, for the different ARLs, this is possible but only before the sonde loading in the ARL trays.
GRUAN Data Processing (GDP) is currently applied to the ARL soundings performed by the GRUAN
stations though the related measurement programs cannot as yet be certified as GRUAN products.
The present analysis has provided a substantive move forwards towards this aim by showing that
performance is broadly comparable to manual launches.
In the last five years, several discussions within and outside the GRUAN community, involving also
the manufactures, allowed to identify a few possibilities to meet the full traceability for the ARLs.
Identified solutions to test are related to two main options:
● Use of a SHC (plus a reference thermometer, such as PT100 sonde) immediately after the
manufacturer GC and prior to loading the sondes;
● Use of reference thermometer and hygrometer within the the ready-to-launch sondes
storage area, as close as possible to the radiosonde sensors, with the optional use of a few
additional thermometers and hygrometers within the storage area to monitor the
uniformity of the temperature and relative humidity within the same area.
Both approaches have advantages and drawbacks. The first allows use of the SHC as a traceable
calibration standard at or around 100 % relative humidity, depending on the solution used in the
SHC. Nevertheless, the proposed two stage procedure can be applied only in advance of the launch
and tests are needed to confirm what was already shown in Section 4 at Sodankylä and Payerne



stations, i.e. a sonde can be launched within a few days from its upload in the ARL without differing
significantly from the SHC collected data.
The second approach can instead continuously monitor the radiosonde during the entire launch
procedure in the storage area and before the sonde tray is moved out to the vessel area for launch,
when temperature and RH within the storage area may rapidly change because of the incoming air
from outside the vessel area. This approach cannot directly use traceable calibration standards but
it must be based on the comparison with reference thermometers and hygrometers calibrated on
a routine and certified basis. In addition, the sonde calibration cannot be monitored at 100 % RH
because the air conditioning system within the ARL keeps stable humidity conditions and cannot be
modified to avoid an impact on the ARL operation efficiency.
For both the approaches above, a customized solution to collect the data and use them in the
generation of a GDP must be found given the constraints of the ARL software which does not allow
extra calibration or comparison values to be collected or saved in the main radiosonde launch files.
It must be noted that at 4 JMA stations, not belonging to GRUAN, the Vaisala ARL is used adopting
a modified setup of the AS15 system including an additional GC based on reference instruments
developed by Vaisala for temperature and humidity, i.e Vaisala HMP155 with HMT333, lodged in a
custom-made chamber. When loading the radiosonde, the JMA specified GC for temperature and
humidity is also performed, in line with JMA's rule for upper air observations, specifying that the
PTU radiosonde sensors should be compared to reference sensors before launch only to confirm
that the difference is within a pre-defined threshold, while reference values are not used for any
correction of the measured profiles. The JMA additional GC is not a traceable calibration standard
and does not allow to perform the 0% RH and 100% RH ground calibration immediately before the
launch. Instead, it can be made when the radiosonde is uploaded in the ARL using a method to save
the measured comparison values.
More details on the JMA specified ground check for temperature and humidity are available at:
https://www.vaisala.com/sites/default/files/documents/RI41-Datasheet-B211322EN.pdf.
The compilation of the table of ARL systems in Appendix A (also the plot in Figure 1) brought home
that it is not easy for users to know which stations are using ARLs.  We recommend that information
on automated launchers (type, start date, end date if appropriate) should be included in the
OSCAR/Surface catalogue.
Other issues which must be considered and solved to provide a GDP from ARLs are related to the
need to supply the manufacturer software with an accurate local pressure measurement and its





height at the launch time. Delays between the actual and the reported launch time from the
software is another issue which is under investigation by GRUAN community.
The GRUAN community is discussing a strategy to achieve the full traceability for the ARL products
and to ascertain if any of the approaches described above can be tested intensively at one or more
sites: unfortunately, many of the GRUAN sites are also operational stations from the Met Services
and from other research institutions and are not readily available for testing. The next step will be
to identify which sites can perform specific tests on the ARL traceability and to collect as many
metadata as possible from all the GRUAN sites to report, in following publications, extensive
statistics validating the results presented in this paper.

**8. Acknowledgements**
Much useful information has been provided by the three manufacturers: Vaisala, Meteomodem and
Meisei. Information on which stations use Meteomodem ARLs was provided by Adrien Ferreira of
Meteomodem in April 2019. Hannu Jauhiainen of Vaisala provided a list of stations using their
Autosonde including several which were not known from the WIS reports. MeteoFrance and several
other National Meteorological Services have also provided information. The Faa'a data discussed in
this manuscript are available at ftp://ftp.lmd.polytechnique.fr/jcdupont/data_m10_gruan_faa and
can be used or cited under the DOI number https://doi.org/10.14768/20181213001.1.

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

**10. APPENDIX A: Table of ARL systems operating around the world**
Table A1: ARL stations shown in Figure 1. For each station, the WMO ID, which is also part of the WIGOS code
(https://oscar.wmo.int/surface), the latitude, the longitude, the country and the period of installation is reported. For
the approximate installation date (year or year-month), the metadata have been collected from different sources (IGRA,
ECMWF, manufacturers, personal communication from scientists and instrument operators). If the last column is empty,
no clear information on the installation period at that station are available. For Vaisala systems the "radiosonde type"
in the reports should indicate if an ARL is being used, but it has been found that this is not always coded correctly.  For
Modem and Meisei systems there is no way for the current code formats to indicate that an ARL has been used.  The
list is ordered according to the WMO ID.

| WMO ID | Latitude | Longitude | Country | Installed |
|---|---|---|---|---|
| 01001 | 70.940 | -8.668 | Norway | Meteomodem 2019-09 |
| 01010 | 69.315 | 16.131 | Norway | Vaisala 2014 |
| 01241 | 63.705 | 9.612 | Norway | Vaisala 2001 |
| 01415 | 58.874 | 5.665 | Norway | Vaisala 2013 |
| 01492 | 59.943 | 10.719 | Norway | Vaisala 1997 |
| 02185 | 65.543 | 22.115 | Sweden | Vaisala 1996 |
| 02365 | 62.532 | 17.436 | Sweden | Vaisala 1994 |



| | | | | |
|---|---|---|---|---|
| 02527 | 57.657 | 12.291 | Sweden | Vaisala 1994 |
| 02591 | 57.671 | 18.345 | Sweden | Vaisala pre-1996 |
| 02836 | 67.366 | 26.631 | Finland | Vaisala 2005-12 |
| 02963 | 60.815 | 23.499 | Finland | Vaisala 1998 |
| 03238 | 55.019 | -1.878 | UK | Vaisala 1999 |
| 03354 | 53.006 | -1.250 | UK | Vaisala 1999 |
| 03882 | 50.891 | 0.317 | UK | Vaisala 2001 |
| 03918 | 54.503 | -6.343 | UK | Vaisala 2002 |
| 03953 | 51.939 | -10.241 | Ireland | Meteomodem 2015 |
| 04018 | 63.975 | -22.588 | Iceland | Vaisala 2006 |
| 04360 | 65.611 | -37.637 | Greenland | Meteomodem 2012 |
| 06610 | 46.813 | 6.943 | Switzerland | Vaisala 2018 |
| 07110 | 48.444 | -4.412 | France | Meteomodem 2016-04 |
| 07145 | 48.770 | 2.020 | France | Meteomodem 2015-04 |
| 07510 | 44.831 | -0.691 | France | Meteomodem 2012-06 |
| 07645 | 43.856 | 4.407 | France | Meteomodem 2011-11 |
| 07761 | 41.918 | 8.792 | France | Meteomodem 2014-06 |
| 08190 | 41.384 | 2.118 | Spain | Meteomodem 2012 |



| 08221 | 40.465 | -3.589 | Spain | Vaisala 2002 |
|---|---|---|---|---|
| 08392 | 39.606 | 2.707 | Spain | Vaisala 2002 |
| 08383 | 37.278 | -6.911 | Spain | Vaisala 2018 |
| 08430 | 38.002 | -1.171 | Spain | Meteomodem 2015 |
| 10035 | 54.527 | 9.550 | Germany | Vaisala 2019-10 |
| 10113 | 53.712 | 7.152 | Germany | Vaisala 2011 |
| 10410 | 51.404 | 6.968 | Germany | Vaisala 2012 |
| 10548 | 50.562 | 10.377 | Germany | Vaisala 2011 |
| 10739 | 48.828 | 9.201 | Germany | Vaisala 2012 |
| 10868 | 48.245 | 11.553 | Germany | Vaisala 2013 |
| 11010 | 48.232 | 14.201 | Austria | Vaisala 2016 |
| 11120 | 47.260 | 11.355 | Austria | Vaisala 2015 |
| 11240 | 46.994 | 15.447 | Austria | Vaisala 2015 |
| 13388 | 43.327 | 21.898 | Serbia | Meteomodem 2015 |
| 14430 | 44.101 | 15.339 | Croatia | Vaisala 1999 |
| 16113 | 44.539 | 7.613 | Italy | Vaisala 1999 |
| 16144 | 44.654 | 11.623 | Italy | Vaisala 1998 |
| 45004 | 22.312 | 114.173 | Hong Kong | Vaisala 2003 |



| | | | | |
|---|---|---|---|---|
| 47155 | 35.170 | 128.573 | S Korea | Vaisala 2001 |
| 47418 | 42.953 | 144.438 | Japan | Vaisala 2010-03 |
| 47600 | 37.391 | 136.895 | Japan | Vaisala 2010-03 |
| 47678 | 33.122 | 139.779 | Japan | Meisei 2010-03 (Vaisala until 2003-06) |
| 47741 | 35.458 | 133.066 | Japan | Vaisala 2010-03 |
| 47778 | 33.45 | 135.757 | Japan | Vaisala 2010-03 |
| 47909 | 28.393 | 129.552 | Japan | Meisei 2007-03 |
| 47918 | 24.337 | 124.165 | Japan | Meisei 2006-03 |
| 47945 | 25.829 | 131.229 | Japan | Meisei 2017-03 (Vaisala until 2005-03) |
| 60018 | 28.318 | -16.382 | Spain | Vaisala 2001 |
| 60096 | 23.705 | -15.930 | Morocco | Meteomodem 2012 |
| 60155 | 33.559 | -7.667 | Morocco | Meteomodem 2014 |
| 61980 | -20.9 | 55.500 | La Reunion | Meteomodem 2018-04 |
| 70026 | 71.287 | -156.763 | USA, Alaska | Vaisala  2010 |
| 70133 | 66.885 | -162.597 | USA, Alaska | Vaisala 2019 |
| 70200 | 64.513 | -165.443 | USA, Alaska | Vaisala 2019 |





| 70219 | 60.780 | -161.838 | USA, Alaska | Vaisala 2018 |
|---|---|---|---|---|
| 70231 | 62.953 | -155.603 | USA, Alaska | Vaisala 2018 |
| 70261 | 64.814 | -147.859 | USA, Alaska | Vaisala 2018 |
| 70273 | 61.175 | -149.993 | USA, Alaska | Vaisala 2018 |
| 70308 | 57.167 | -170.22 | USA, Alaska | Vaisala 2018 |
| 70326 | 58.678 | -156.647 | USA, Alaska | Vaisala 2019 |
| 70350 | 57.750 | -152.494 | USA, Alaska | Vaisala 2015 |
| 70361 | 59.503 | -139.66 | USA, Alaska | Vaisala 2018 |
| 70398 | 55.043 | -131.571 | USA, Alaska | Vaisala 2018 |
| 71964 | 60.733 | -135.097 | Canada | Vaisala 1997 |
| 78897 | 16.260 | -61.510 | Gaudeloupe | Meteomodem 2015 |
| 81405 | 4.830 | -52.370 | French Guyana | Meteomodem 2012-09 |
| 89859 | -74.624 | 164.232 | Antarctic (S. Korea) | Vaisala 2014 |
| 91592 | -22.27 | 166.450 | New Caledonia | Meteomodem 2016-06 |



| 91938 | -17.55 | -149.6 | Tahiti | Meteomodem 2018-10 |
|---|---|---|---|---|
| 94170 | -12.678 | 141.921 | Australia | Vaisala 1998 |
| 94302 | -22.241 | 114.097 | Australia | Vaisala 1997 |
| 94312 | -20.373 | 118.632 | Australia | Vaisala 1998 |
| 94332 | -20.679 | 139.488 | Australia | Vaisala 1998 |
| 94430 | -26.613 | 118.536 | Australia | Vaisala 1998 |
| 94510 | -26.414 | 146.257 | Australia | Vaisala 1998 |
| 94637 | -30.784 | 121.454 | Australia | Vaisala 2000 |
| 94653 | -32.13 | 133.698 | Australia | Vaisala 1999 |
| 94659 | -31.156 | 136.805 | Australia | Vaisala 2000 |
| 94711 | -31.484 | 145.897 | Australia | Vaisala 1997 |
| 94776 | -32.793 | 151.836 | Australia | Vaisala 2002 |
| 94821 | -37.748 | 140.775 | Australia | Vaisala 2010 |
| 94995 | -31.542 | 159.077 | Australia | Vaisala 2010 |
| 95527 | -29.49 | 149.847 | Australia | Vaisala 1999 |
| 96996 | -12.189 | 96.834 | Australia | Vaisala 1997 |






Table A2: Additional ARL systems not transmitting data through the WIS in 2019 or used only for tests and short
campaign (not shown in Figure 1).  The ARL from 08160 was relocated to 08383.

| Identifier | Latitude | Longitude | Country | Installed |
|---|---|---|---|---|
| POT (GRUAN) | 40.600 | 15.725 | Italy | Vaisala 2004 |
| 08160 | 41.660 | -1.000 | Spain | Vaisala 2005 to 2016 |
| 72402 (test) | 37.930 | -75.480 | USA | Vaisala 2014 Meteomodem 2017 |
| 71461 (test) | 55.810 | -117.890 | Canada | Vaisala 2016 Meteomodem 2017 |
| 10141 (test) | 53.650 | 10.117 | Germany | Vaisala 2016 |
