# Peer review of "Use of automatic radiosonde launchers to measure temperature and humidity profiles from the GRUAN perspective"

_Atmospheric Measurement Techniques, 2019_

## Referee Comment (RC1) · Anonymous Referee #1 · 16 Mar 2020

This is a good and relevant paper to publish for Atmospheric Measurement Techniques because the paper's first steps towards demonstrating reliability of Automatic Radiosonde Launchers (ARLs) hold important news for the worldwide radiosonde community. Some revision of typos and unclear language is advisable; I'll provide my list of 'technical corrections' in support of this.

---

## Referee Comment (RC2) · Anonymous Referee #1 · 17 Mar 2020

I list here my technical corrections as well as suggestions for corrections to improve understanding. I list the corrections in order of manuscript line number, referring to the line numbering of this manuscript version.

30 'Sondakyla'. Please spell the station name the same ('Sodankylä') throughout the manuscript.

173 Last word should be 'RS41-SG' instead of 'RS41-SGP'.

271-272 Trappes station latitude, longitude is listed as '48.46N,0.20E, 168 m asl'. This is inconsistent with the manuscript table A1 entry for 07145: '48.770 , 2.020 ' and with WMO OSCAR/Surface for Trappes reporting '48.774444 N, 2.0097222222 E, 167 m

asl. Please correct or explain clearly if the manual and automated Trappes stations have different positions.

282 A suggestion: I failed to notice the mention of 'how new' the mentioned 'new system' is. In the summary (line 873), Vaisala and MeteoModem are selected as 'the two most mature systems at present'. If you mention the (lack of) maturity of Meisei ARL here in section 2.3, the summary will be easier to read.

297-298 After the '.' something is missing from the sentence to make ' he GC performs before the sonde loading' make sense.

312-450 A suggestion: Insert a table defining the terms 'effective flights', 'successful launches' and 'successful flights' according to MeteoSwiss and MeteoFrance respectively. And be clear in the text when which is referred to.

383 Figure 5: Please replace with a mature figure without confusing red text and red error marks :-)

396 Exchange 'Effective flights' with 'Successful flights'. At least if it is correctly guessed that the '470 successful flights' mentioned in line 395 are 'successful' according to the MeteoSwiss standard mentioned in line 396-397 by 'Effective flights according to MeteoSwiss standards are'?

401 Please rephrase this information after the comma: 'with a limited use of spare sondes due to the failure of scheduled launches (4 %)'. At least, I do not understand the intended message. Unless it is something to do with the ARL having limited access to spares, because somebody needs to be around to refill the ARL for the number of spares to be without limit? But before the comma 'manual launches' are mentioned last, and therefore the sentence after the comma, should refer to those. But according to table 4 'percentage of spare ' is not available for the manual flights at Payerne. In short: I do not understand the intended meaning, please rephrase.

404-407 Please rewrite, to make the sentences easy to understand, unambiguous

and consistent with the rest of the paper. I.e. How should this sentence in line 404-406 be understood: 'the Meteomodem ARL Robotsonde in Trappes has realized 1908 successful flights, out of a total of 1956 successful flights according to MeteoFrance standards'? Who 'realized' the remaining 48 'flights' out of the 'total of 1956 successful flights'? Manned personnel? If so, please mention in the text the existence of 'some flights after manual launch' at Trappes during the 2016-2018, automated period. Or, should the sentence rather be understood as the '1908 successful flights' being successful according to MeteoSwiss standards? If so, please write it out, to avoid confusion like mine :-)

421 Please clear up this apparent inconsistency regarding the number of scheduled and/or successful flights at Trappes in 2018: After the period the text reads: 'For the 578 flights performed during 2018'. But the reader expects Trappes to have made at least 723 successful launches in 2018 (99,1% of 'two launches per day (line 394) for 365 days') and at least 716 successful flights (99% of 723). Why was only 578 flights performed in 2018?

428 Table 4 caption: Please add text clarifying if 'percentage of successful flights' is defined as 'percentage of successful flights out of scheduled flights' or 'percentage of successful flights out of successful launches' or if it is not necessarily specified precisely how the respondents defined this.

431 After the komma, please replace 'sondes' with 'balloons' in the text 'the sondes bursting'.

477-479 The sentence starting after the period in line 477: Please rephrase to make this important and educational author assumption clearer, e.g. write 'The results are shown in Figure 7 assuming' instead of 'The results shown in Figure 7 assume'.

498 delete either the second or third word ('by' or 'for') in the sentence 'collected by for GRUAN station launching'

524 Replace reference to 'Figure 4' with reference to 'Figure 9' (!)

530-531 Delete text '; from left to the right, the time period increases from 1 to more than 5 days' as it is not consistent with figure 9.

534 Suggestion for clarity: Replace text 'a similar study to' with something like 'another kind of GC study than'.

544-553 Delete text (it is the repeated text of lines 534-543)

632 I suggest adding the word 'eventual' (or 'potential') in front of the words 'positive influence' so that the sentence starts as 'A more interesting comparison to show the eventual positive influence of automation on the burst altitude'. 'Eventual' (or 'potential') because the Trappes analysis apparently shows an immediate set back in availability after automation (>99% percentage of successful manual flights in 2012 drops to ~95,5% successful automated flights (figure 6) in 2016). But a very important message of the Trappes example is how the performance improved over time and especially after the provider in November 2016 made the improvement of (line 411-415) 'a flexible cover which assures that during the storage the balloon is less exposed to contact with the air-conditioned environment. This seems to reduce the effects of drier air on the balloon and improve its performance in terms of burst altitude (standard deviation of burst altitude is reduced after the installation of the cover – not shown)'.

635-636 Before the word 'respectively' in line 636, please reverse the order of the ascent velocities to be consistent with (according to figure 14) the order 'ARL and manual launches' have in the beginning of line 635.

640 I suggest replacing 'significantly' with 'much' (or something like it) or else mention which test was used to determine significance. The two distributions in figure 14, left panel look different to me :-)

642-643 I suggest for clarity, please repeat/insert here more details on 'the operational organization' as it might not be clear to every reader, that they should recall the

potential beneficial switch to Totex balloons as well as other things mentioned in line 410-415.

648 I suggest to ask MeteoFrance for their own explanation of the apparent difference in burst height distributions (Figure 14 right panel) of the old manned and the new automated station and include it in the analysis.

602-720 Please note the inconsistent section numbering in line 677 and revise it and the rest of the numbering/headers of section 5.

680 Delete misleading words 'the left panel of'.

695 End of sentence is missing. And maybe also a verb is missing.

722 Please correct station position for Faa'a so that it is consistent and easy to iden-tify ('French Polynesia, 28.34S, 16.32E' is inconsistent). Is the stations referred to as 'Faa'a' the same as table A1 entry WMO id 91938 having coordinates -17.55 , -149.6? If so it would be helpful to readers to confirm this in the text by saying so or by mention-ing the WMO station name 'TAHITI-FAAA' or the WIGOS station id 0-20000-0-91938 along with the correct position.

732-751 I suggest to move this to 'section 3 Technical performance' to highlight this, because this information on very misleading observations in the lower 50-100 m is very important, interesting and general (e.g. it's not only Faa'a since ECMWF notes 'some reports' from 'stations') including how one of the suppliers recently implemented remedying software at some stations.

762-764 Text inconsistent with the figure 19 it describes. To remedy please e.g. in line 762 replace 'the difference' with 'the mean difference'. In line 764 replace 'correspond-ing mean difference' with 'corresponding standard deviation'.

792-795 Inconsistent conclusion starting with the words 'Applying the Wilcoxon Rank-Sum Test'. Please rephrase to be consistent so as to EITHER state 'the two distribu-tions of burst altitudes are not significantly different' OR 'ARL does lead to improve-

ments in the balloon burst altitude'. The text says both, which is inconsistent.

853-855 Please add units in the figure caption text.

902 I strongly suggest completing the picture by mentioning here also the observed mean deviation of 2 K in temperature and 4% in RH in the first 50-100 meters of the Faa'a series.

---

## Referee Comment (RC3) · Anonymous Referee #2 · 16 Apr 2020

This paper presents an evaluation of the performance of radiosonde atmospheric observations made using Automatic Radiosonde Launchers (ARL). Comparisons from different stations between the ARL and manually launched radiosonde data and with meteorological analyses are presented to quantify the performance of the ARL approach to radiosonde measurements. The evaluation is put into the context of the global ensemble of radiosonde stations running ARLs. The paper provides evaluation and relevant technical details for ARLs from 3 radiosonde manufacturers. The paper concludes that overall performance of the ARL launches is comparable to manual launches, although there are differences that would benefit from further investigation.

Overall this is a well-organized and substantial paper on a relevant topic for weather prediction and climate studies. There are some technical corrections which should be made before publication. These are itemized below:

Line 40: the abbreviation O-B (observation-background) should be defined here

Line 71: what is meant by "basic" equipment here? Does this mean rudimentary, or limited, or less capable (eg lower precision versus equipment in a conventional laboratory environment)? Please clarify.

Line 78: "progress" could be replaced with "innovation" for a better style

Line 165: "5% RH for" instead of "5% RH or"

Line 191-192: How accurate is this procedure? Eg, how high does the temperature need to rise before the RH is effectively zero relative to the desired calibration threshold?

Figure 2 top panel: the small white words are very hard to read. Can you enlarge the font?

Line 254: is there a reference available for the Rotronic HC2A-S probe?

Line 284: it is unclear what "a maximum number of 40 sondes adjustable" means, does this mean there is maximum of up to 40 sondes, and the maximum can be adjusted by the user?

Lines 304-307: the meaning of this is a little unclear; is it that at this time, Meisei considers the information proprietary, or that additional information is at a preliminary/developing state?

Figure 5: I'm concerned the font will be illegible due to small size when this is formatted for publication

Line 410: why was the switch made to Totex? Is there a cost or supply or reliability
reason the switch wasn't made earlier?

Figure 5: why is the % successful flights based only on 2018, when there are >2.5 years of previous data? Was the equipment/equipment operation not optimized until 2018?

Lines 518-520: more (but brief) information on what the Wilcoxon Rank Sum Test, and why it was used, would be good here. It's better described later in the text (eg around line 692)

Line 524: right panel of Figure 4 should say "shows" (grammar)

Lines 525-527: although the test data for Sondakyla are not shown, can you briefly summarize the outcome?

Figure 9: the noise in the profile plots makes them somewhat hard to grasp and interpret; would it be possible to replace by bar graphs binned by altitude for 3-5 altitude bins?

Lines 544-553: this text is repeated, please delete

Figure 18: why does the difference grow rapidly with height from 5-15 km and then stabilize? Is there just more variability in upper troposphere winds vs lower troposphere, and then calmer winds in stratosphere?

lines 788-795: is the probability close between the daytime and nighttime launches? It looks like the daytime launches differ more than the nighttime launches between ARL and manual.

Figure 6: what does the abbreviation "nb" mean?

---

## Author Comment (AC3) · 5 May 2020

The authors appreciate the positive opinion of the reviewer #1 on the manuscript and provides a point-to-point reply to the technical corrections reported in the additional comment uploaded by reviewer (RC2).

---

## Author Response (AR1)

**Reply to the technical corrections by the anonymous referee #1**

The authors of the manuscript gratefully acknowledge the positive opinion on the manuscript and the helpful comments provided by the anonymous reviewer #1, which aim at increasing clarity and readability of the manuscript itself. In the new version of the manuscript, which shall be uploaded once the AMT discussion stage will be closed, all the technical suggestions provided by the reviewer will be included.

In particular, the authors want to provide an immediate feedback to reply to the most interesting points raised by the reviewer. The latter are reported in the following with the authors' replies (preceded by the letter "R")

Line 30: 'Sondakyla'. Please spell the station name the same ('Sodankylä') throughout the manuscript.

R: This is an unexpected mistake due the conversion of the manuscript in pdf.

Lines 271-272: Trappes station latitude, longitude is listed as '48.46N,0.20E, 168 m asl'. This is inconsistent with the manuscript table A1 entry for 07145: '48.770, 2.020' and with WMO OSCAR/Surface for Trappes reporting '48.774444 N, 2.0097222222 E, 167 m asl. Please correct or explain clearly if the manual and automated Trappes stations have different positions.

R: The coordinates reported for Trappes station are those declared by the station operators for GRUAN, please cheek also https://www.gruan.org/network/sites.

Line 312-450 The reviewer commented that: "A suggestion: Insert a table defining the terms 'effective flights', 'successful launches' and 'successful flights' according to MeteoSwiss and MeteoFrance respectively. And be clear in the text when which is referred to."

R: In the new version of the manuscript, two footnotes with the definition of 'successful flights' have been included in the considered page.

Figure 5: Please replace with a mature figure without confusing red text and red error marks.

R: The authors apologize for the confusing text and marks: the mistakes have been removed in the new version of the manuscript.

Please clear up this apparent inconsistency regarding the number of scheduled and/or successful flights at Trappes in 2018: After the period the text reads: 'For the 578 flights performed during 2018'. But the reader expects Trappes to have made at least 723 successful launches in 2018 (99,1% of 'two launches per day (line 394) for 365 days') and at least 716 successful flights (99% of 723). Why was only 578 flights performed in 2018?

R: Yes, the reviewer is right and "578 flights" is a mistake. In the new version of the manuscript, the number of flights has been correctly reported (716).

Formattato: Normale, A destra, Bordo:Superiore: (Nessun bordo), Inferiore: (Nessun bordo), A sinistra: (Nessun bordo), A destra: (Nessun bordo), Tra : (Nessun bordo), Tabulazioni: 8.5 cm, Centrato + 17 cm, A destra, Posizione:Orizzontale: A sinistra, Rispetto a: Colonna, Verticale: In linea, Rispetto a: Margine, Intorno

404-407 Please rewrite, to make the sentences easy to understand, unambiguous and consistent
with the rest of the paper. I.e. How should this sentence in line 404- 406 be understood: 'the
Meteomodem ARL Robotsonde in Trappes has realized 1908 successful flights, out of a total of 1956
successful flights according to MeteoFrance standards'? Who 'realized' the remaining 48 'flights'
out of the 'total of 1956 success- ful flights'? Manned personnel? If so, please mention in the text
the existence of 'some flights after manual launch' at Trappes during the 2016-2018, automated
period. Or, should the sentence rather be understood as the '1908 successful flights' being
successful according to MeteoSwiss standards? If so, please write it out, to avoid confusion like mine
:-)
**R: Yes, the reviewer is right and the paragraph has been re-elaborated to clarify as follows: "the**
**Meteomodem ARL Robotsonde in Trappes has realized 1908 successful flights, according to**
**MeteoFrance standards, out of a total of 1956. For each of the remaining 48 flights, a spare**
**automatic launch was performed which fulfilled the requirements of Meteofrance."**
428 Table 4 caption: Please add text clarifying if 'percentage of successful flights' is defined as
'percentage of successful flights out of scheduled flights' or 'percentage of successful flights out of
successful launches' or if it is not necessarily specified precisely how the respondents defined this."
**R: The reported percentage is the percentage "of successful flights out of successful launches".**
**This is now clearly reported in the text using a footnote in the considered page.**
Lines 642-643 I suggest for clarity, please repeat/insert here more details on 'the operational
organization' as it might not be clear to every reader, that they should recall the potential beneficial
switch to Totex balloons as well as other things mentioned in line 410-415.
**R: As suggested by the reviewer, the authors added a few more details in this paragraph about**
**the operational organization, which is carried out under a joint effort between Meteomodem and**
**MeteoFrance the overall management of the site (including loading and type of balloon, balloon**
**inflation without human contact, preparation of radiosonde before flights for calibration, both**
**with ground-check, meteorological shelter and saturated chamber, system check-up, etc).**
648 I suggest to ask MeteoFrance for their own explanation of the apparent difference in burst
height distributions (Figure 14 right panel) of the old manned and the new automated station and
include it in the analysis.
**R: Figure 14 shows (1) a thinner and sharper data frequency distribution for the automatic system**
**than for the manual that can be related with a more homogeneous balloon inflation (automatic**
**inflation, same method, constant gas flow, more stable temperature), and (2) a higher peak**
**occurrence frequency that can be related with the use of better balloon and with less human**
**contact.**
**The text reported in the new version of the manuscript is the following: "The comparison reveals**
**that the burst altitude (Figure 14, right panel) is generally higher for the ARL than for the manual**
**launches, likely due to use of different balloons and the more limited human contact with balloon**
**itself. ARL frequency distribution has also a more peaked distribution that can be related with a**
**more homogeneous balloon inflation (automatic inflation, same method, constant gas flow, more**
**stable temperature)."**

**Formattato:** Normale, A destra, Bordo:Superiore: (Nessun bordo), Inferiore: (Nessun bordo), A sinistra: (Nessun bordo), A destra: (Nessun bordo), Tra : (Nessun bordo), Tabulazioni: 8.5 cm, Centrato + 17 cm, A destra, Posizione:Orizzontale: A sinistra, Rispetto a: Colonna, Verticale: In linea, Rispetto a: Margine, Intorno

Please correct station position for Faa'a so that it is consistent and easy to identify ('French Polynesia, 28.34S, 16.32E' is inconsistent). Is the stations referred to as 'Faa'a' the same as table A1 entry WMO id 91938 having coordinates -17.55, -149.6? If so it would be helpful to readers to confirm this in the text by saying so or by mentioning the WMO station name 'TAHITI-FAAA' or the WIGOS station id 0-20000-0-91938 along with the correct position.

**R: Yes, the reviewer is right. The correct position of Faa'a site is Latitude: 17°33.298' S, Longitude: 149°36.876' W (17.63S, 149.84W in decimal degrees). The WIGOS station ID is 0-20000-0-91938. All this information has been reported in the new version of the manuscript ensuring consistency across the sections.**

732-751 I suggest to move this to 'section 3 Technical performance' to highlight this, because this information on very misleading observations in the lower 50-100 m is very important, interesting and general (e.g. it's not only Faa'a since ECMWF notes 'some reports' from 'stations') including how one of the suppliers recently implemented remedying software at some stations.

**R: According to the reviewer's suggestion, the paragraph at lines 732-751 has been moved to the section 3, where the Technical performance of the ARL systems are discussed.**

**Formattato:** Normale, A destra, Bordo:Superiore: (Nessun bordo), Inferiore: (Nessun bordo), A sinistra: (Nessun bordo), A destra: (Nessun bordo), Tra : (Nessun bordo), Tabulazioni: 8.5 cm, Centrato + 17 cm, A destra, Posizione:Orizzontale: A sinistra, Rispetto a: Colonna, Verticale: In linea, Rispetto a: Margine, Intorno

**Reply to the anonymous referee #2**

**The authors of the manuscript gratefully acknowledge the positive opinion on the manuscript and the helpful comments provided also by the anonymous reviewer #2, which aim at further increasing clarity of the manuscript itself, with a particular focus on the figure and on the outcome of the applied statistical tests. In the new version of the manuscript, all the technical suggestions provided by the reviewer have been included.**

**Nevertheless, here the authors provide a point-to-point reply to the reviewer suggestions and comments. The authors' response is reported below, always preceded by the letter "R" and in bold.**

Line 40: the abbreviation O-B (observation-background) should be defined here.

**R: observation-minus-background has been defined in the abstract.**

Line 71: what is meant by "basic" equipment here? Does this mean rudimentary, or limited, or less capable (eg lower precision versus equipment in a conventional laboratory environment)? Please clarify.

**R: at Line 71 "basic" means limited, for example very often the manual launches are performed using a more basic technology for the control of balloon filling than those available in the Automatic Radiosounding Launchers. To avoid confusion the sentence has been modified removing the second part: "*During the preparation and launch phase, many circumstances may interfere with the smooth operation of radiosoundings such as undertaking launches at night, harsh meteorological conditions for balloon train preparation, if any, and safe handling when using hydrogen as balloon gas, and last but not least the risk of errors/mishandling by the operators.*".**

Line 78: "progress" could be replaced with "innovation" for a better style

**R: done.**

Line 165: "5% RH for" instead of "5% RH or"

**R: done.**

Line 191-192: How accurate is this procedure? Eg, how high does the temperature need to rise before the RH is effectively zero relative to the desired calibration threshold?

**R: According to the information shared by the manufacturer, the outcome of an uncertainty study of the RS41 relative humidity measurements after ground preparation showed an uncertainty (k = 2) of 0.5–2 % RH at a temperature of 20°C and RH ranging from 0 to 100 % [1], and laboratory test results support the stated uncertainties [2].**

**[1] Vaisala: Vaisala Radiosonde RS41 Measurement Performance White Paper. Ref. B211356EN-A © Vaisala, 2013.**

**[2] Vaisala: Comparison of Vaisala Radiosondes RS41 and RS92 White Paper. Ref. B211317EN – B © Vaisala, Helsinki, Finland, 2014. Vaisala: Vaisala Radiosonde RS41 White Paper – Ground Check Device R141. Ref. B211539EN-A © Vaisala, 2015.**

**A reference to the two documents above has been added to the manuscript.**

Figure 2 top panel: the small white words are very hard to read. Can you enlarge the font?

**R: during the writing phase of the manuscript this issue already came out; nevertheless, this picture was kindly provided by Vaisala and should be, according got them, the only one available to describe the size of the interior sectors of the Vaisala Autosonde AS41. As a consequence, the authors apologize but they would prefer to leave Figure 2 in its current shape**

Line 254: is there a reference available for the Rotronic HC2A-S probe?

**R: https://www.rotronic.com/en/hc2a-s.html. This link has been added at the corresponding line.**

Line 284: it is unclear what "a maximum number of 40 sondes adjustable" means, does this mean there is maximum of up to 40 sondes, and the maximum can be adjusted by the user?

**R: The word adjustable has been removed.**

Lines 304-307: the meaning of this is a little unclear; is it that at this time, Meisei considers the information proprietary, or that additional information is at a preliminary/developing state?

**R: Meisei, as well JMA, did not run any parallel sounding to investigate and improve the performance of their system, which is currently commercialized; therefore, a final assessment of the system performance cannot be made available yet. Despite the limited number of information made available for this manuscript by Meisei, the authors agreed on the importance to report in this work all the information on all the Automated Radiosounding Launchers available on the market.**

Figure 5: I'm concerned the font will be illegible due to small size when this is formatted for publication

**R: In the new manuscript version, the diagram in Figure 5 has been replaced with a 300dpi version, without modifying its current shape. The printing of the Figure appears to authors readable.**

Line 410: why was the switch made to Totex? Is there a cost or supply or reliability reason the switch wasn't made earlier?

**R: Since September 2015, HWOYEE 600 balloon were replaced by Totex TX1000 at Trappes station. This change is simply explained by the result of a call for tenders made by MeteoFrance for the renewal of the balloon purchasing framework at the end of 2013.**

Figure 5: why is the % successful flights based only on 2018, when there are >2.5 years of previous data? Was the equipment/equipment operation not optimized until 2018?

**Formattato:** Normale, A destra, Bordo:Superiore: (Nessun bordo), Inferiore: (Nessun bordo), A sinistra: (Nessun bordo), A destra: (Nessun bordo), Tra : (Nessun bordo), Tabulazioni: 8.5 cm, Centrato + 17 cm, A destra, Posizione:Orizzontale: A sinistra, Rispetto a: Colonna, Verticale: In linea, Rispetto a: Margine, Intorno

**R: The authors suspect that the comment provided by the reviewer refers to Table 5 and not to**
**Figure 5. If this is the case, The % successful flights in the presented statistics refer to one year**
**only (2018) to consider a period of the same length as that considered for the statistics presented**
**for the Payerne Vaisala ARL. This study can be considered fully is representative of the 2.5 years**
**of data collected with the Meteomodem ARL.**
Lines 518-520: more (but brief) information on what the Wilcoxon Rank Sum Test, and why it was
used, would be good here. It's better described later in the text (eg around line 692).
**R: At lines 518-520, the following text has been added: "The Wilcoxon Rank Sum Test is a non-**
**parametric test of the null hypothesis that it is equally likely that a randomly selected value from**
**one population will be less than or greater than a randomly selected value from a second**
**population. If the null hypothesis is rejected, that there is evidence that the medians of the two**
**populations differ. In this study, the Wilcoxon Rank Sum Test has been used instead of the Z-test**
**due to its robustness in case of a small observations sample (i.e. small number of parallel**
**launches) and to avoid assumptions on the underlying data distribution (e.g. data distribution**
**skewed or non-normal)."**
Line 524: right panel of Figure 9 should say "shows" (grammar)
**R: Fixed.**
Lines 525-527: although the test data for Sondakyla are not shown, can you briefly summarize the
outcome?
**R: The additional test data for Sodankylä, mentioned in the manuscript, refers to a very long**
**storage-time and the test was made in a similar manner to the one shown in Figure 10. In this**
**case, the radiosondes used for the test were not launched in parallel to the manual launches as**
**done instead for the dataset shown in Figure 9. The test was carried on performing a first ground**
**check, then the sonde was left on a tray of the ARL for up to one month period and after that**
**another ground check was made. The ground check showed almost identical values even after a**
**long tray time.**
**As a consequence, to avoid misunderstandings, the authors decided to remove the sentence at**
**lines 525-527.**
Figure 9: the noise in the profile plots makes them somewhat hard to grasp and interpret; would it
be possible to replace by bar graphs binned by altitude for 3-5 altitude bins?
**R: In the new version of the manuscript, a bar plot has replaced the line plot. The text has been**
**refined accordingly.**
Lines 544-553: this text is repeated, please delete
**R: done.**
Figure 18: why does the difference grow rapidly with height from 5-15 km and then stabilize? Is
there just more variability in upper troposphere winds vs lower troposphere, and then calmer winds
in stratosphere?

**Formattato:** Normale, A destra, Bordo:Superiore: (Nessun bordo), Inferiore: (Nessun bordo), A sinistra: (Nessun bordo), A destra: (Nessun bordo), Tra : (Nessun bordo), Tabulazioni: 8.5 cm, Centrato + 17 cm, A destra, Posizione:Orizzontale: A sinistra, Rispetto a: Colonna, Verticale: In linea, Rispetto a: Margine, Intorno

**R: In Figure 18, it is shown the horizontal distance calculated for the balloons of the 21 parallel**
**soundings performed at Faa'a station. The horizontal distance of two parallel soundings is mainly**
**determined in troposphere by advection, turbulence, the time difference between the two**
**launches and the balloon filling which determines the ascending speed. The latter is very**
**important to determine the balloon motion if combined with the effect of horizontal winds. The**
**distance may also increase quickly depending on the combination of the described factors.**
**In lower stratosphere, winds are a laminar flow (i.e. there is small turbulence) and this combined**
**also with a slower ascending speed due to the balloon deformation at lower pressure does not**
**increase the balloon distance as in the troposphere.**
lines 788-795: is the probability close between the daytime and nighttime launches? It looks like the
daytime launches differ more than the nighttime launches between ARL and manual.
**R: The probability calculated for the balloon burst altitude dataset at Faa's station is obtained**
**applying the Wilcoxon Rank Sum Test to night (11 launches) and daytime data (10 launches)**
**together. Beyond the small size of the dataset, the objective of the test was to compare the overall**
**performance for the entire dataset. If we separate daytime and night time, considering the**
**smaller size of the two datasets, the median values show a larger difference during daytime than**
**at night time. Nevertheless, the results of a statistical test would be more affected by the size of**
**the dataset and the authors prefer to apply the Wilcoxon Rank Sum Test on the entire dataset.**
**The text at lines 788-795 has been slightly modified to clarify.**
Figure 6: what does the abbreviation "nb" mean?
**R: "nb" stands for "number". To avoid misunderstandings, this has been specified in the figure**
**caption.**

**Formattato:** Normale, A destra, Bordo:Superiore: (Nessun bordo), Inferiore: (Nessun bordo), A sinistra: (Nessun bordo), A destra: (Nessun bordo), Tra : (Nessun bordo), Tabulazioni: 8.5 cm, Centrato + 17 cm, A destra, Posizione:Orizzontale: A sinistra, Rispetto a: Colonna, Verticale: In linea, Rispetto a: Margine, Intorno

[revised manuscript text omitted]

available on the Vaisala website (https://www.vaisala.com).

**Formattato:** Normale, A destra, Bordo:Superiore: (Nessun bordo), Inferiore: (Nessun bordo), A sinistra: (Nessun bordo), A destra: (Nessun bordo), Tra : (Nessun bordo), Tabulazioni: 8.5 cm, Centrato + 17 cm, A destra, Posizione:Orizzontale: A sinistra, Rispetto a: Colonna, Verticale: In linea, Rispetto a: Margine, Intorno

[Figure]

[Figure]

[Figure]

Figure 2: Schematics of the VAISALA Autosonde AS41 system in its most recent configuration (top panel), and photo of
the Autosonde system AS15 (bottom panel) operational at the Finnish Meteorological Institute GRUAN site in Sodankylä
(WIGOS station identifier=0-20000-0-02836, 67.34 °N, 26.63 °E, 179 m a.s.l., see Vaisala 2018,
https://www.vaisala.com/sites/default/files/documents/AUTOSONDE%20AS41%20Datasheet%20B211636EN-
A_2%20pages.pdf)).

**Formattato:** Normale, A destra, Bordo:Superiore: (Nessun bordo), Inferiore: (Nessun bordo), A sinistra: (Nessun bordo), A destra: (Nessun bordo), Tra : (Nessun bordo), Tabulazioni: 8.5 cm, Centrato + 17 cm, A destra, Posizione:Orizzontale: A sinistra, Rispetto a: Colonna, Verticale: In linea, Rispetto a: Margine, Intorno

Table 1: Autosonde AS41 technical data (Vaisala, 2018)

[revised manuscript text omitted]

This section aims to provide a classification of the main challenges met by the stations which have operated ARLs over several years and to assess the technical performance of the ARLs compared to manual launches. The section is built upon the feedback provided by the GRUAN sites in response to a survey for the collection of ARL information. Most of the ARLs at GRUAN sites are from Vaisala (thus the analysis is not representative of Meisei and Meteomodem systems due to the very limited

Formattato: Normale, A destra, Bordo:Superiore: (Nessun bordo), Inferiore: (Nessun bordo), A sinistra: (Nessun bordo), A destra: (Nessun bordo), Tra : (Nessun bordo), Tabulazioni: 8.5 cm, Centrato + 17 cm, A destra, Posizione:Orizzontale: A sinistra, Rispetto a: Colonna, Verticale: In linea, Rispetto a: Margine, Intorno feedback available for these systems). Given the small sample size, this is presented qualitatively rather than quantitatively and it is anonymised. Examples of technical performance in the field are then provided for a Vaisala and a Meteomodem ARL operating the most recent updated version of the respective manufactured systems (at Payerne and Trappes stations).

A conceptual diagram to represent a generic ARL is provided in Figure 5: each ARL can be schematically divided into 4 areas as follows:

- the operator's area, where the operators can manage the system, prepare radiosondes and balloons to be uploaded and where the station reception and processing units are located;
- the ready-to-launch sondes storage area, built around the ARL rotating trays, where most of the automated technologies are implemented to allow a completely unmanned launch;
- the launching vessel area, where the balloon is filled and becomes ready for the launch;
- external area, where all the ancillary instruments, such as the weather station and GNSS antenna, are located along with gas tanks.

For each area, the weakest points identified from the GRUAN sites operating an ARL are:

- in the operator's area, most of the issues are related to the not infrequent failure of power supply system or of the air conditioning system, often related to a major failure of the power supply at the measurement station itself. This represents a particular weakness in the use of ARLs in remote areas where power supply is generally less stable, and where logically the ARL might be an obvious choice. A few sites also reported issues in the software and logic controllers;
- the ready-to-launch sonde storage area is assessed as the most efficient part of ARLs, where few issues reported. he most critical issue identified in this area is the infrequent failure of the air compressor;
- the launching vessel area is where the balloon is filled and launched and where, therefore, we have a high exposure to many environmental factors like harsh climate, dust, animals, etc., which can strongly affect a successful launch also with later effects to the balloon and early burst. Several issues were raised by the stations related to challenges in the balloon inflation process, failure of balloon presence sensor allowing launch of under-inflated balloons, gas tubes bent and frozen gas hoses, balloon blocked on the tray, failure of the rams which open vessel cover doors (this concerns Vaisala or Meisei, and not Meteomodem ARL). Other issues noted were delays in launch detection time compared to the actual launch time, and occasional break of the radiosonde string at launch (for Meisei);

Formattato: Normale, A destra, Bordo:Superiore: (Nessun bordo), Inferiore: (Nessun bordo), A sinistra: (Nessun bordo), A destra: (Nessun bordo), Tra : (Nessun bordo), Tabulazioni: 8.5 cm, Centrato + 17 cm, A destra, Posizione:Orizzontale: A sinistra, Rispetto a: Colonna, Verticale: In linea, Rispetto a: Margine, Intorno

[revised manuscript text omitted]

**Formattato:** Normale, A destra, Bordo:Superiore: (Nessun bordo), Inferiore: (Nessun bordo), A sinistra: (Nessun bordo), A destra: (Nessun bordo), Tra : (Nessun bordo), Tabulazioni: 8.5 cm, Centrato + 17 cm, A destra, Posizione:Orizzontale: A sinistra, Rispetto a: Colonna, Verticale: In linea, Rispetto a: Margine, Intorno

| | | | | |
|---|---|---|---|---|
| 07145 | 48.770 | 2.020 | France | Meteomodem 2015-04 |
| 07510 | 44.831 | -0.691 | France | Meteomodem 2012-06 |
| 07645 | 43.856 | 4.407 | France | Meteomodem 2011-11 |
| 07761 | 41.918 | 8.792 | France | Meteomodem 2014-06 |
| 08190 | 41.384 | 2.118 | Spain | Meteomodem 2012 |
| 08221 | 40.465 | -3.589 | Spain | Vaisala 2002 |
| 08392 | 39.606 | 2.707 | Spain | Vaisala 2002 |
| 08383 | 37.278 | -6.911 | Spain | Vaisala 2018 |
| 08430 | 38.002 | -1.171 | Spain | Meteomodem 2015 |
| 10035 | 54.527 | 9.550 | Germany | Vaisala 2019-10 |
| 10113 | 53.712 | 7.152 | Germany | Vaisala 2011 |
| 10410 | 51.404 | 6.968 | Germany | Vaisala 2012 |
| 10548 | 50.562 | 10.377 | Germany | Vaisala 2011 |
| 10739 | 48.828 | 9.201 | Germany | Vaisala 2012 |
| 10868 | 48.245 | 11.553 | Germany | Vaisala 2013 |
| 11010 | 48.232 | 14.201 | Austria | Vaisala 2016 |
| 11120 | 47.260 | 11.355 | Austria | Vaisala 2015 |
| 11240 | 46.994 | 15.447 | Austria | Vaisala 2015 |

**Formattato:** Normale, A destra, Bordo:Superiore: (Nessun bordo), Inferiore: (Nessun bordo), A sinistra: (Nessun bordo), A destra: (Nessun bordo), Tra : (Nessun bordo), Tabulazioni: 8.5 cm, Centrato + 17 cm, A destra, Posizione:Orizzontale: A sinistra, Rispetto a: Colonna, Verticale: In linea, Rispetto a: Margine, Intorno

[revised manuscript text omitted]

**Pagina 41: [6] Eliminato**      **Fabio Madonna**      **08/05/20 11:09:00**

[Figure]

**Pagina 43: [7] Eliminato**      **Fabio Madonna**      **08/05/20 11:09:00**

---

## Author Response (AR2)

**Response to the Editor's comment**

**The authors thank the Editor for the additional comments provided and related to some final inconsistencies in the text compared to the recommendations provided by the two anonymous reviewers. All the points raised by the Editor ahs been fixed in the final version. Below we provided the text adjustments in track changes.**

**The authors want to stress that now there is full consistency between Figure 3 and the table in the Appendix, while a comment is provided to clarify the removal of the repeated text at the Lines 544-553 of the manuscript version 1.**

[revised manuscript text omitted]

**Commentato [FM1]:** According to the reviewer #2, these lines were repeated, not this is not the case anymore. The mistaken duplication of the text was removed.

[revised manuscript text omitted]